**Effect of precipitation seasonality on annual oxygen isotopic composition in the area of spring persistent rain in southeastern China and its palaeoclimatic implication**

**Haiwei Zhang[1, 2], Hai Cheng[1, 3], Yanjun Cai[2, 1], Christoph Spötl[4], Ashish Sinha[5], Gayatri Kathayat[1] and Hanying Li[1]**

[1]Institute of Global Environmental Change, Xi'an Jiaotong University, Xi'an 710054, China

[2]Institute of Earth Environment, Chinese Academy of Sciences, State Key Laboratory of Loess and Quaternary Geology, Xi'an 710061, China

[3]Department of Earth Sciences, University of Minnesota, Minneapolis, Minnesota 55455, USA

[4]Institute of Geology, University of Innsbruck, Innsbruck 6020, Austria

[5]Department of Earth Science, California State University Dominguez Hills, Carson, California 90747, USA

**Correspondence**: zhanghaiwei@xjtu.edu.cn

**Key Points:**

- Precipitation seasonality in the SPR region in Southeast China is different from that in other monsoon regions of China.

- The ENSO modulates the precipitation seasonality in the SPR region and influences the oxygen isotopic composition of rainfall.

- Understanding the spatial differences in seasonal precipitation are key to a robust interpretation of speleothem records in the monsoon region of China.

**Abstract.** This study examines the seasonality of precipitation amount and $\delta^{18}O$ over the monsoon region of China (MRC). We found that the precipitation amount associated with the East Asian summer monsoon (EASM) in the spring persistent rain (SPR) region is equivalent to that of the non-summer monsoon (NSM). The latter contributes ~50% to amount-weighted annual $\delta^{18}O$ values, in contrast with other areas in the MRC, where the $\delta^{18}O$ of annual precipitation is dominated by EASM precipitation. Interannual relationships between the ENSO index, simulated $\delta^{18}O$ data from IsoGSM, and seasonal precipitation amount in the SPR region were also examined. We found that on interannual timescales, the seasonality of precipitation amount (EASM/NSM ratio) was modulated by ENSO and primarily influences the variability of amount-weighted annual precipitation $\delta^{18}O$ values in the SPR region, although integrated regional convection and moisture source and transport distance may also play subordinate roles. During El Niño (La Niña) phases, less (more) EASM and more (less) NSM precipitation leading to lower (higher) EASM/NSM precipitation amount ratios results in higher (lower)

amount-weighted annual precipitation $\delta^{18}O$ values and, consequently, in higher (lower) speleothem $\delta^{18}O$ values. Characterizing spatial differences in seasonal precipitation is, therefore, key to correctly interpreting speleothem $\delta^{18}O$ records from the MRC.

**Key words**: Precipitation seasonality, Oxygen isotopes, Spring persistent rain, East Asian summer monsoon, ENSO, Back-trajectory analysis


**1 Introduction**

Summertime rainfall over the MRC is largely associated with the East Asian summer monsoon (EASM) (Figure 1a) (Ding, 1992). However, a significant portion of annual rainfall in southeastern China also occurs during springtime (i.e., from March to mid-May), known as the spring persistent rain (SPR).
The SPR occurs mostly south of the middle and lower reaches of the Yangtze River (~ 24 °N to 30 °N, 110 °E to 120 °E) (Figure 1b) and is a unique synoptic and climatic phenomenon in East Asia (Tian and Yasunari, 1998; Wan and Wu, 2007, 2009). The SPR is another rainy period before the Meiyu rain period in early summer and covers the region from southeastern China to the south of Japan. It has long been debated whether the SPR marks the onset of EASM. Ding (1992) called SPR an "early summer rainy
season" and considered it as a part of the summer monsoon rainfall (Ding et al., 1994). He et al. (2008) suggested that the SPR marks the establishment of the East Asian subtropical monsoon, which is considered as a component of the EASM. Other studies suggest that the SPR is unrelated to EASM rainfall and consider it as extension of winter atmospheric circulation (Tian and Yasunari, 1998; Wan and Wu, 2009). Wang and Lin (2002) proposed that the SPR over southeastern China is not a part of the
EASM, because the large-scale circulation and rain-bearing systems differ from those associated with summer monsoon rainfall. Tian and Yasunari (1998) suggested that the SPR is the effect of the land-sea thermal contrast and is unrelated to topographical effects, as there is a coherent increase of the spring rain from southeastern China to southern Japan. Wan et al. (2008a, 2009) proposed that the formation of SPR is primarily influenced by the mechanical and thermal forcing of the Tibetan Plateau. Without this
topographic element, the SPR rain belt would not exist. Climatic factors from the mid-high latitudes and the tropics also influence the interannual variability of the SPR (Feng and Li, 2011; Wu and Kirtman, 2007; Wu and Mao, 2016).

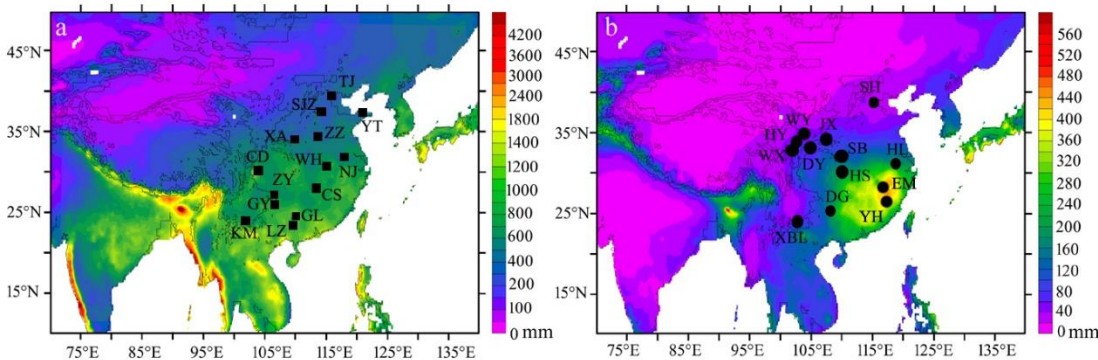

**Figure 1**. Overview map showing the spatial distribution of seasonal precipitation amount in China and locations mentioned in this study. (a) Regional mean EASM (May-September) precipitation amount (mm/month) in China from 1951 to 2007. The black squares represent the locations of GNIP stations (TJ-Tianjin, YT-Yantai, SJZ-Shijiazhuang, XA-Xi'an, ZZ-Zhengzhou, NJ-Nanjing, WH-Wuhan, CS-Changsha, CD-Chengdu, ZY-Zunyi, GY-Guiyang, GL-Guilin, LZ-Liuzhou, KM-Kunming, details can be found in Table 1). (b) Regional mean SPR (March-April) precipitation amount (mm/month) in China from 1951 to 2007. The SPR is obvious in southeastern China from about 24 °N to 30 °N, and from 110 °E to 120 °E. The black circles represent the locations of caves with published stalagmite records (SH-Shihua cave (Li et al., 2017), HL-Hulu cave (Wang et al., 2001), SB-Sanbao cave (Cheng et al., 2016), HS-Heshang cave (Hu et al., 2008), DG-Dongge cave (Yuan et al., 2004), XBL-Xiaobailong (Tan et al., 2017) cave, WY-Wuya cave (Tan et al., 2014), DY-Dayu cave (Tan et al., 2009), WX-Wanxiang cave (Zhang et al., 2008), HY-Huangye cave (Tan et al., 2010), EM-E'mei cave (Zhang et al., 2018), YH-Yuhua cave (Jiang et al., 2012)). Precipitation data source: APHRODITE (Asian Precipitation-Highly-Resolved Observational Data Integration Towards Evaluation of Water Resources, APHRO_MA_V1101R2 product, (21)) (Yatagai et al., 2009).

Although considerable emphasis has been placed on understanding the causes and mechanisms of SPR, little is known about its precipitation $\delta^{18}O$ ($\delta^{18}O_p$) variability and about the mechanisms that produce this variability (Tan, 2016; Zhang, 2014). Based on rainfall monitoring data from eight sites in the EASM region, Tan et al. (2016) found that, in 2012 AD, the spring rainfall amount was equivalent to the summer rainfall but their $\delta^{18}O_p$ values were different. They suggested that the seasonal $\delta^{18}O_p$ variability is affected by the changes of moisture source but not the precipitation amount variations. Huang et al. (2017) and Wu et al. (2015) studied the $\delta^{18}O_p$ variability at the Changsha station located in the SPR region and its relationship with the El Niño-Southern Oscillation (ENSO) (Figure 1a), but did not focus on the $\delta^{18}O_p$ variability of SPR. A better understanding of the $\delta^{18}O_p$ variability in the SPR region on seasonal to interannual time scales, however, is crucial for a robust interpretation of the oxygen isotopic data of Chinese speleothems from this region (e.g., Cai et al., 2015; Cheng et al., 2009, 2016; Wang et al., 2001, 2008; Yuan et al., 2004; Zhang et al., 2008). Several mechanisms including the amount

effect, moisture source/transport distance, integrated regional convection, winter temperature, and precipitation seasonality have been shown to influence the $\delta^{18}O_p$ and speleothem $\delta^{18}O$ to various degrees and at different timescales across the MRC (Cai et al., 2018; Caley et al., 2014; Cheng et al., 2016; Clemens et al., 2010; Dayem et al., 2010; Maher, 2008, 2016; Maher and Thompson, 2012; Pausata et al., 2011; Tan, 2016; Zhang et al., 2018). The SPR region is located within the area of the EASM and its rainy season includes both summertime monsoon rainfall and SPR (Wan and Wu, 2009). Therefore, the factors that influence the $\delta^{18}O_p$ in this region are likely complex. The aim of this study is to examine this climate-$\delta^{18}O$ proxy relationship during the instrumental period. To this end, we compare the seasonal variations of precipitation amount and $\delta^{18}O_p$ in the SPR region with other regions of the MRC and discuss the interannual variations of precipitation amount and $\delta^{18}O_p$ in the SPR region and their relationship with the large-scale oceanic-atmospheric circulation.

**2 Data and Methods**

2.1 Meteorological data

A daily gridded precipitation dataset for 1951-2007 was obtained from APHRODITE (Asian Precipitation-Highly-Resolved Observational Data Integration Towards Evaluation of Water Resources, APHRO_MA_V1101R2 product, (21)) (Yatagai et al., 2009). The regional mean SPR (March-April) and EASM (May-September) precipitation amount in China from 1951 to 2007 are shown in Figure 1, which was exported based on this dataset using the free software Ferret (https://ferret.pmel.noaa.gov/Ferret).

Monthly precipitation datasets of 160 meteorological stations in China for the period 1951-2014, obtained from the National Climate Center (http://ncc.cma.gov.cn), were used to characterize the percentage of spring (March-April) and EASM (May-September) precipitation amount relative to the annual precipitation amount in China.

Monthly mean $\delta^{18}O_p$ and precipitation amount data from meteorological stations across the MRC were obtained from the Global Network for Isotopes in Precipitation (GNIP) (http://www.iaea.org/) (Table 1 and Figure 1a). The monthly mean $\delta^{18}O_p$ data are used to compare the seasonal to interannual variation of $\delta^{18}O_p$ in the MRC. The stations near the coast from the southeastern region of the MRC (Fuzhou, Haikou, Hongkong, Guangzhou) were excluded, because their precipitation amount and $\delta^{18}O_p$ are significantly influenced by typhoons in summer and autumn. Changsha station is the only GNIP station in the SPR region.

2.2 $\delta^{18}O_p$ data from IsoGSM simulations

IsoGSM is a water isotope-permitting general circulation model (Yoshimura et al., 2008). We use the product of IsoGSM nudged toward the NCEP/NCAR Reanalysis 2 (Kanamitsu et al., 2002) atmosphere and forced with observed sea-surface temperatures (SST) and sea ice data (Yoshimura et al.,

2008). A detailed description of the model setup can be found in Yoshimura et al. (2008) and Yang et al.
(2016). IsoGSM can reproduce reasonably well monthly variabilities of precipitation and water vapor
isotopic compositions associated with synoptic weather cycles. In order to verify the reliability of the
simulated data from the IsoGSM, we first cross-compare the data from GNIP Changsha station with those
from the IsoGSM during 1988-1992. The good replication indicates that both the precipitation amount
and the $\delta^{18}O$ data from the IsoGSM simulation are consistent (Supplementary Figure 1).

2.3 Ocean-atmosphere circulation index

ENSO plays an important role in governing the climatic variation in the MRC (e.g., Feng and Hu,
2004; Xue and Liu, 2008; Zhou and Chan, 2007). We used the Southern Oscillation Index (SOI)
multivariate ENSO Index (MEI) to calculate the correlations between the phases of ENSO, the $\delta^{18}O_p$ and
the seasonal precipitation amount. The SOI is defined as the normalized pressure difference between
Tahiti and Darwin. Negative and positive values of SOI represent El Niño and La Niña events,
respectively. The data were obtained from the Australian Government Bureau of Meteorology
(http://www.bom.gov.au/climate/current). The MEI is based on six oceanic-atmospheric variables (sea-
level pressure, zonal and meridional components of the surface wind, SST and total cloudiness fraction
of the sky) over the tropical Pacific, and is used to examine the role of ENSO in influencing the rainfall
over the MRC. The MEI is defined as the first principal component of above six variable fields. Therefore,
it provides a more complete description of the ENSO phenomenon than a single variable ENSO index
such as the SOI or Niño 3.4 SST (Wolter and Timlin, 2011). Positive and negative values of MEI
represent El Niño and La Niña events, respectively. The data were obtained from the website of the Earth
System Research Laboratory, National Oceanic & Atmospheric Administration (NOAA)
(http://www.cdc.noaa.gov/people/klaus.wolter/MEI). Tropical Pacific SST show a La Niña phase during
the period from May 1988 to May 1989 and an El Niño phase during the period from May 1991 to June
1992. Therefore, we define 1988-1989 as La Niña years (1988 is the developing year and 1989 is the
decaying year of the La Niña event) and 1991-1992 as El Niño years (1991 is the developing year and
1992 is the decaying year of the El Niño event) in this paper.

The Arctic Oscillation (AO) can also influence the climate and precipitation over the MRC (Gong
et al., 2001, 2011; He et al., 2017; Li et al., 2014). It was suggested that a warmer winter in East Asia (a
positive winter AO value) is associated with increased winter rainfall in southern parts of East Asia, and
a positive spring AO is followed by increased rainfall in southern China but decreased rainfall in the
lower valley of Yangtze River (He et al., 2017). We also calculated the correlation between the AO index
and the seasonal rainfall amount in our study area. The data were downloaded from the website of NOAA
(https://www.cpc.ncep.noaa.gov/products/precip/CWlink/daily_ao_index/ao.shtml#forecast).

2.4 Back-trajectory and moisture source contribution calculation

The Hybrid Single-Particle Lagrangian Integrated Trajectory model (HYSPLIT) (Stein et al., 2015) was used to perform air mass back trajectory calculations for the GNIP Changsha station during the period 1988-1992. In order to qualitatively assess the moisture source regions and transport paths for rainy season precipitation only air mass back trajectories for precipitation-producing days were used. Trajectories were initiated four times daily (at UTC 00:00, 00:60, 12:00, and 18:00) during precipitating days (>1 mm precipitation/day) and their air parcel was released at 1500 m above ground level and moved backward by winds for 120 hours (5 days). To identify the moisture uptake locations along the back trajectories during 1988-1992, we followed the method described in Sodemann et al. (2008) and Krklec et al. (2018). Two criteria (i.e., a more conservative threshold of positive gradient in specific humidity (0.2 g/kg within 6 h), and initial relative humidity of more than 80%) were used to identify moisture uptake locations along the back trajectories. Following the methodology of Krklec et al. (2018), we calculated the contributions of moisture uptake locations en route to the precipitation in GNIP Changsha station and provide a map showing the percentage of moisture uptake contributing to Changsha precipitation during La Niña (1988-1989) and El Niño (1991-1992) years. A grid of 0.5×0.5 degrees was used for the computation of the moisture uptake locations.

**3 Results**

3.1 Proportions of SPR, EASM and NSM precipitation over MRC

We calculated the mean ratios of spring (March-April) to annual precipitation (spring/annual) and EASM (May-to-September) to annual precipitation (EASM/annual) ratios for the period 1951-2014. Figure 2a shows that the mean percentage of spring/annual in southeastern China (about $20°N$ to $33°N$, $107°E$ to $122°E$, red rectangle in Figure 2a), which range from 10-25% and 0-10% in northern (about $33°N$ to $53°N$, $100°E$ to $134°E$, pink polygon in Figure 2a) and southwestern regions of the MRC (about $20°N$ to $33°N$, $90°E$ to $107°E$, green polygon in Figure 2a), respectively. The Jiangxi and the eastern Hunan Provinces, the core regions of the SPR, show the highest mean percentage of spring/annual within the MRC (20-25%). which is consistent with the results from the previous studies (Tian and Yasunari, 1998; Wan and Wu, 2009). Figure 2b shows that the mean percentage of EASM/annual is 40-70% in southeastern China and 70-95% in other regions of the MRC. Conversely, the mean percentage of non-summer monsoon precipitation to annual precipitation (NSM/annual) is 30-60% in southeastern China and 5-30% in other regions of the MRC and reaches the maximum in the SPR region (45-60%). This indicates that the proportion of EASM precipitation (40-55%) is nearly equivalent to the proportion of NSM precipitation (45-60%) in the SPR region.

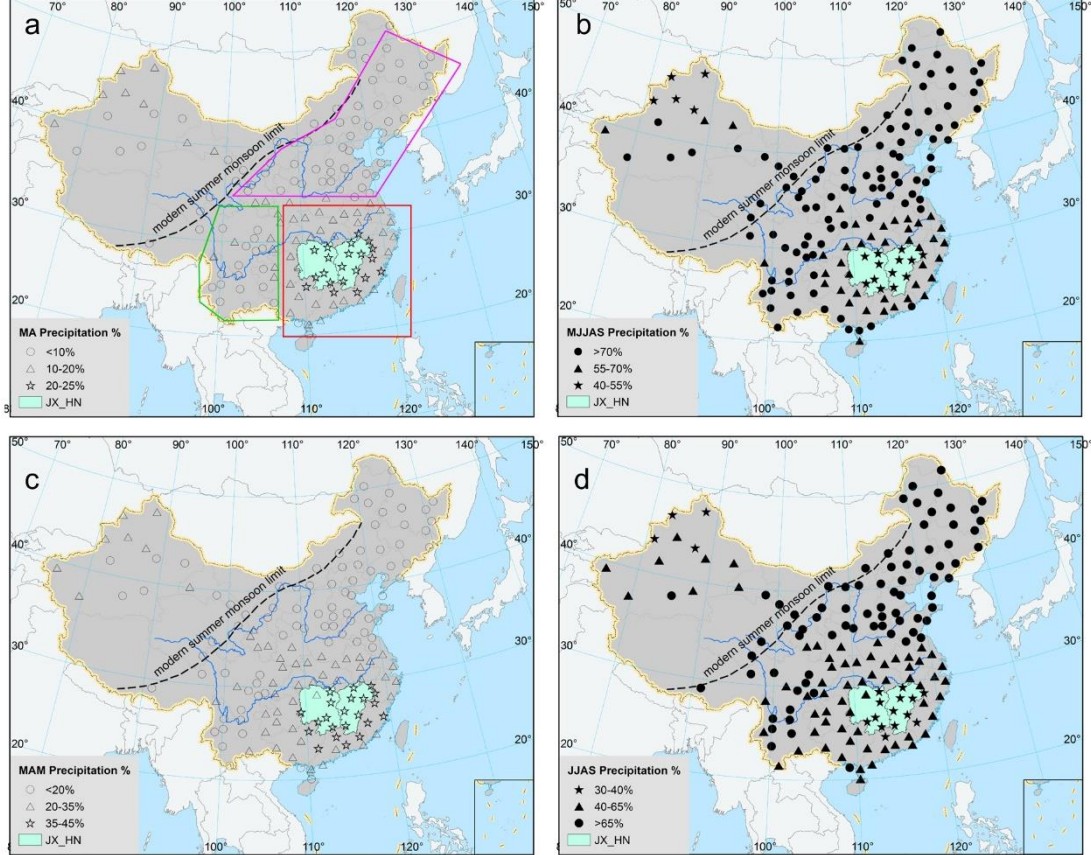

**Figure 2**. The percentage of spring (a, March to April) and EASM (b, May to September) precipitation amount relative to the annual precipitation amount in China. Figures c and d are similar to a and b, except that spring precipitation is shown from March to May in c and EASM precipitation between June and September in d. The Jiangxi and Hunan Provinces (JX_HN) are highlighted in jade color. The monthly precipitation data (1951-2014) from 11 meteorological stations (Jiujiang, Guixi, Nanchang, Guangchang, Ji'an, Ganzhou, Changsha, Yueyang, Hengyang, Chenzhou, Xinning) in the Jiangxi Province and the eastern Hunan Province were used to examine the relationship between ocean-atmospheric circulation, precipitation amount and $\delta^{18}O$ in the SPR region. The red, pink and green polygons in panel a indicate southeastern, northern and southwestern regions of the MRC, respectively.

Usually, the SPR period lasts from March to mid-May (Wan and Wu, 2009) and the EASM period lasts from mid-May to September (Wang and Lin, 2002), however, the onset/retreat time of SPR and EASM and their intensities vary in different years (Zhou and Chan, 2007). The EASM starts late (late May to early June) and tends to be weaker during El Niño years (Huang et al., 2012) and EASM precipitation amount over Southeast China is reduced when the SPR starts later and lasts longer time (until late May) (Wan et al., 2008a). Therefore, if we define the March-to-May precipitation as SPR and the June-to-September precipitation as EASM in El Niño years, the mean percentage of SPR/annual in the SPR region is 35-45% (Figure 2c), the mean percentage of EASM/annual is only 30-40% (Figure 2d),

and the mean percentage of NSM/annual is 60-70%. However, in other regions of the MRC, the mean percentage of EASM/annual (65-90%) is still much higher than the mean percentage of NSM/annual (10-35%) (Figures 2c and d). Conversely, during La Niña years, the March-April and May-to-September precipitation should be defined as SPR and EASM precipitation, respectively (Figures 2a and b). Therefore, the distribution of EASM vs. NSM precipitation amount in the SPR region is distinctly different from that in other regions of the MRC, and the ratio of EASM/NSM precipitation amount in the SPR region might be influenced by ENSO. We discuss this in detail in the section 4.2.

3.2 Seasonal precipitation $\delta^{18}O_p$ and amount over the MRC

We compared the seasonal variations of precipitation amount and $\delta^{18}O_p$ in the SPR region with those in other regions of the MRC by using data from the GNIP stations. According to the spatial distribution of EASM precipitation as discussed in section 3.1, we assigned Zhengzhou, Xi'an, Yantai, Shijiazhuang, and Tianjin GNIP stations to northern region of the MRC, Kunming, Guiyang, Zunyi, and Chengdu GNIP stations to southwestern region of the MRC, and Changsha, Guilin, Liuzhou, Nanjing, and Wuhan GNIP stations to southeastern China. Only the Changsha GNIP station is located in the SPR region (Table 1 and Figure 1a).

**Table 1**. GNIP stations used for the comparison of the seasonal precipitation amount and $\delta^{18}O_p$ in the MRC.

| Category | Sites |
| --- | --- |
| Northern region of the MRC | Zhengzhou (34°43′12″N, 113°39′00″E) |
| | Xi'an (34°18′00″N, 108°55′48″E) |
| | Yantai (37°31′48″N, 121°24′00″E) |
| | Shijiazhuang (38°1′60″N, 114°25′01″E) |
| | Tianjin (39°6′00″N, 117°10′01″E) |
| Southwestern region of the MRC | Kunming (25°1′00″N, 102°40′59″E) |
| | Guiyang (26°34′60″N, 106°43′01″E) |
| | Zunyi (27°41′60″N, 106°52′48″E) |
| | Chengdu (30°40′12″N, 104°1′12″E) |
| Southeastern region of the MRC | Changsha (28°11′60″N, 113°4′01″E) |
| | Guilin (25°4′12″N, 110°4′48″E) |
| | Liuzhou (24°21′00″N, 109°24′00″E) |
| | Nanjing (32°10′48″N, 118°10′48″E) |
| | Wuhan (30°37′12″N, 114°7′48″E) |

The seasonal variation of $\delta^{18}O_p$ in the MRC is consistently related to the onset, advancement, and retreat the EASM. The $\delta^{18}O_p$ values decrease in May as the summer monsoon starts (Figure 3). The $\delta^{18}O_p$ values are relatively low during the monsoon season (June-August) (Figure 3), because of the long-distance transport of water vapor from the distal Indian Ocean to the MRC. Along this pathway, progressive rainout associated with regional convection leads to more negative $\delta^{18}O_p$ values via Rayleigh
distillation (Baker et al., 2015; He et al., 2018; Liu et al., 2010; Moerman et al., 2013; Tan, 2014). The $\delta^{18}O_p$ values become progressively higher as the EASM withdraws in September (Figure 3). From October to next April, the $\delta^{18}O_p$ values are rather high (Figure 3), resulting from the short-distance transport of water vapor from the western Pacific Ocean or local moisture recycling and local convection (He et al., 2018; Moerman et al., 2013; Tan et al., 2016; Wu et al., 2015). The low $\delta^{18}O_p$ values in winter
in northern region of the MRC are caused by the temperature effect, but it is less important because of its small contribution to the amount-weighted mean annual precipitation $\delta^{18}O$ ($\delta^{18}O_w$) (Cheng et al., 2012). Therefore, the seasonal $\delta^{18}O_p$ values over the MRC show a broadly consistent pattern reaching a maximum in March-April and a minimum in July-August in the MRC with the exception of low winter $\delta^{18}O_p$ values in northern region of the MRC.

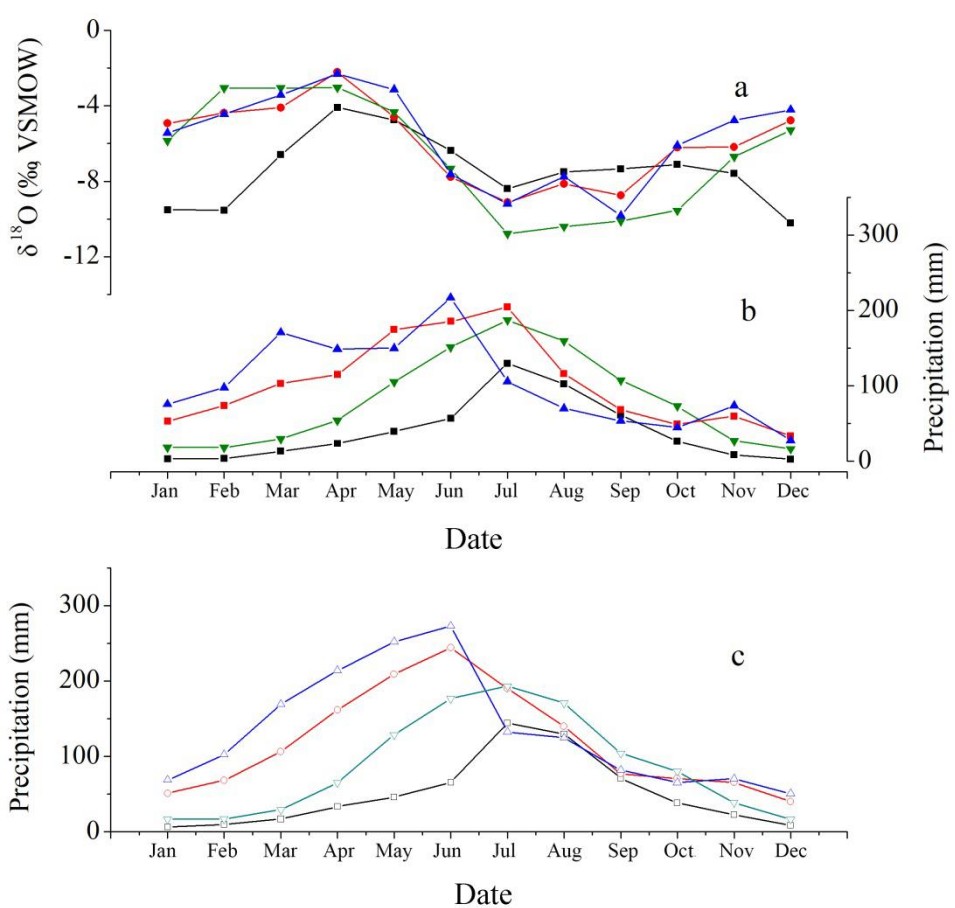

**Figure 3.** Monthly mean $\delta^{18}O_p$ (a) and precipitation amount (b) data from GNIP stations in northern region of the MRC (black lines), southwestern region of the MRC (green lines), southeastern China (red

lines), and the SPR region (blue lines, Changsha station) as grouped in Table 1 (the spatial distribution of the GNIP stations shown in Figure 1a). (c) Monthly mean precipitation data from the meteorological stations closest to the GNIP stations in northern region of the MRC (black lines), southwestern region of the MRC (jade lines), southeastern China (red lines), and the SPR region (blue lines).

Given that there are only a few years of data from those GNIP stations, we obtained the mean monthly precipitation amount from the nearest meteorological station to each GNIP station in the MRC for the period 1951-2014 (Figure 3c). Both datasets show that the seasonal variation of precipitation amount in southeastern China, especially in the SPR region, is different from that in other regions of the MRC (Figures 3b and c). The precipitation amount in March and April before the onset of EASM is high over Southeast China. It is even higher than the summer monsoon precipitation amount in June, July and August in other regions of the MRC. In the SPR region, the summer monsoon precipitation amount in July-August is much smaller than the precipitation in March-April. However, in other regions of the MRC, the summer monsoon precipitation in July-August is the highest of the whole year.

3.3 Moisture source contribution to precipitation in Changsha station

We identified the moisture uptake locations along the back trajectories and calculated their contributions to the precipitation at the GNIP Changsha station during EASM and NSM seasons in a La Niña phase (1988-1989) with low $\delta^{18}O_p$ anomalies and in an El Niño phase (1991-1992) with high $\delta^{18}O_p$ anomalies (Figure 4). The results show that the moisture uptake locations and contributions during the EASM season are similar between El Niño and La Niña phases as well as those during the NSM season. During the EASM season, the moisture sources are mainly from South China Sea-South China, the Bay of Bengal-Indochina Peninsula and the Indian Ocean, while the remaining ones are from North China-western Pacific (Figures 4a and c). In previous studies researchers mainly focused on the variations in moisture source during the EASM season (Baker et al., 2015; Cai et al., 2017; Tan, 2014). In this study, however, we also analyzed the back-trajectories during the NSM season because NSM precipitation contributes ~50% to the annual precipitation in the SPR region. It shows that the NSM moisture sources originate from South China Sea and South China, the remaining ones are driven from local evaporation. Compared to the moisture sources during the EASM season very few moisture sources are indicated for the Bay of Bengal-Indochina Peninsula and the Indian Ocean during the NSM season.

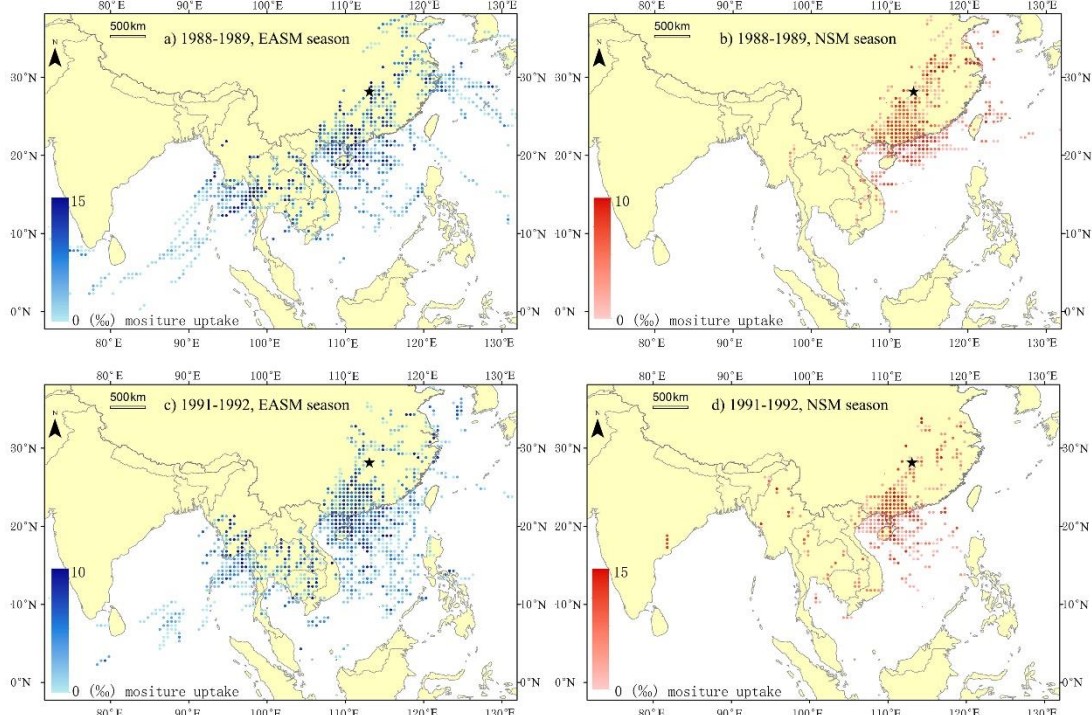

**Figure 4**. Seasonal distribution of moisture uptake contributing to Changsha precipitation in El Niño and La Niña years. (a) and (b) show the moisture source uptake locations and their contribution to precipitation during EASM and NSM seasons in a La Niña phase (1988-1989), respectively; (c) and (d) are the same as (a) and (b) but for an El Niño phase (1991-1992). The black star indicates the Changsha GNIP station.

**4 Discussion**

4.1 Amount-weighted mean annual precipitation $\delta^{18}O$

In principle, the amount-weighted mean annual precipitation $\delta^{18}O$ ($\delta^{18}O_w$) can be calculated from the sum of monthly weighted isotopic values divided by the total amount of precipitation as:

$$\delta^{18}O_w=(P_{Jan}\times\delta^{18}O_{Jan}+P_{Feb}\times\delta^{18}O_{Feb}+\ldots\ldots P_{Dec}\times\delta^{18}O_{Dec})/(P_{Jan}+P_{Feb}+\ldots\ldots P_{Dec}) \quad \text{Eq. (1)}$$

Based on the characteristics of the precipitation amount and $\delta^{18}O$ during EASM and NSM seasons in the MRC, equation (1) can be written in the following mode:

$$\delta^{18}O_w \approx (P_{EASM\text{-}mean}\times\delta^{18}O_{EASM\text{-}mean}+P_{NSM\text{-}mean}\times\delta^{18}O_{NSM\text{-}mean})/(P_{EASM\text{-}mean}+P_{NSM\text{-}mean})$$

$$=EASM\%\times\delta^{18}O_{EASM\text{-}mean}+NSM\%\times\delta^{18}O_{NSM\text{-}mean} \quad \text{Eq. (2)}$$

where $P_{EASM\text{-}mean}$ and $P_{NSM\text{-}mean}$ are the mean precipitation amounts of EASM and NSM, $\delta^{18}O_{EASM\text{-}mean}$ and $\delta^{18}O_{NSM\text{-}mean}$ are the mean values of EASM and NSM precipitation, and EASM% and NSM% are the mean percentages of the EASM and NSM precipitation amounts, respectively.

Therefore, we can consider that the $\delta^{18}O_w$ is controlled by both precipitation amount and $\delta^{18}O_p$ during the EASM and NSM seasons in the MRC. Given the relationship between monthly precipitation amount and $\delta^{18}O_p$ in the MRC (Figure 3) we find that: 1) In northern and southwestern regions of the MRC $\delta^{18}O_w$ are mainly controlled by the amount and $\delta^{18}O$ of EASM precipitation, because the precipitation amount of the EASM with rather low $\delta^{18}O_p$ values accounts for 70% of the annual precipitation and the NSM precipitation is only a small contribution to the $\delta^{18}O_w$ (less than 30%). 2) In Southeast China, especially in the SPR region, the precipitation amount of the NSM with rather high $\delta^{18}O_p$ values even exceeds that of the EASM with rather low $\delta^{18}O_p$ values, and it also has an important effect on $\delta^{18}O_w$. Hence, $\delta^{18}O_w$ in the SPR region is affected by both EASM and NSM precipitation. In addition, except for the effect of the seasonal distribution of precipitation amount, the seasonal $\delta^{18}O$ itself also attributes to the $\delta^{18}O_w$, which is related, among others, to the variations in integrated regional convection and moisture source and transport distance (Cai et al., 2018; Baker et al., 2015; Huang et al., 2017; Tan et al., 2016).

In order to separate the influences of precipitation seasonality and monthly $\delta^{18}O_p$, we used the decomposition method used by Liu and Battisti (2015) and Cai and Tian (2016) to evaluate the role of changes in precipitation seasonality ($\delta^{18}O_{ps}$; assuming that the monthly precipitation $\delta^{18}O_p$ in El Niño years 1988-1989 is the same as that in La Niña years 1991-1992). We then calculated the difference between precipitation $\delta^{18}O_w$ in El Niño years and La Niña years and the change in precipitation $\delta^{18}O$ ($\delta^{18}O_{iso}$; method is similar to that for calculating $\delta^{18}O_{ps}$ but assuming that the monthly precipitation amount is the same). The results for the Changsha station indicate that the difference in precipitation $\delta^{18}O_w$ between El Niño years (1988-1989) and La Niña years (1991-1992) (i.e., El Niño minus La Niña) is 2.7‰, $\delta^{18}O_{ps}$ is 1.3‰, and $\delta^{18}O_{iso}$ is 1.3‰. These results imply that the difference in $\delta^{18}O_w$ between El Niño and La Niña conditions reflects the differences of both the $\delta^{18}O_p$ and the precipitation seasonality.

Tan (2014) suggested that positive (negative) $\delta^{18}O_w$ anomalies during El Niño (La Niña) phases reflect more (less) water vapor originating from the nearby South China Sea and the Western Pacific Ocean (characterized by rather high $\delta^{18}O_p$ values) relative to the remote Indian Ocean (showing comparable low $\delta^{18}O_p$ values). By using the HYSPLIT model, however, Cai et al. (2017) demonstrated that the moisture sources vary little between years with relatively high and low $\delta^{18}O$ values (corresponding to El Niño and La Niña years) in the EASM region; hence EASM precipitation is primarily derived from the Indian Ocean, while the Pacific Ocean moisture is a minor contributor. This is consistent with our results (Figure 4). In addition, by using a Lagrangian precipitation moisture source diagnostic, Baker et al. (2015) suggested that the moisture uptake area in the Pacific Ocean does not differ significantly between summer and winter and is thus a minor contribution to monsoonal

precipitation; changes in moisture transport, however, may impact the $\delta^{18}O$ variation of EASM precipitation. Dayem et al. (2010) also proposed that several processes (e.g., source regions, transport distance and types of precipitation) contribute to the $\delta^{18}O_p$ variation. We found that the moisture sources in the Bay of Bengal-Indochina Peninsula and the Indian Ocean were less important during the NSM season compared to the EASM season (Figure 4). The moisture uptake area in the EASM season does not differ significantly between El Niño and La Niña years nor in the NSM season. Their contributions to the whole precipitation in El Niño and La Niña years, however, are different (Figure 4). The variation in moisture source during the EASM period, in some extent, might contribute to changes in $\delta^{18}O_w$, but it is not the main factor. We also emphasize the effect of NSM precipitation amount on $\delta^{18}O_w$ in the SPR region and we made an attempt to analyze the relationship between the seasonal precipitation amount and $\delta^{18}O_w$ with ENSO phase on interannual timescale in the next section.

4.2 Interannual variation of precipitation amount and $\delta^{18}O_w$ over the SPR region influenced by ENSO

The ENSO is a coupled oceanic-atmospheric phenomenon controlling the interannual variation in precipitation amount and $\delta^{18}O$ over southeastern China (e.g., Feng and Hu, 2004; He et al., 2018; Huang et al., 2017; Moerman et al., 2013; Tan et al., 2014; Xue and Liu, 2008; Yang et al., 2016). Our analysis of the 1988-1992 data from the Changsha GNIP station suggests that the mean value of $\delta^{18}O_w$ (-6.73‰) in La Niña years (1988-1989) is significantly more negative than during El Niño years (1991-1992; -4.11‰). However, there is no significant variation in the annual precipitation amount between La Niña and El Niño years (Figure 5a). The difference of $\delta^{18}O_w$ between La Niña and El Niño phases cannot be explained by variations in annual precipitation amount. This is consistent with the analyses based on instrumental meteorological data (Huang et al., 2017; Tan, 2014) and climate simulations (Yang et al., 2016). Previous studies showed that during El Niño years, the EASM is generally weak and the integrated regional convection decreases in the EASM region, thereby leading to higher $\delta^{18}O_p$ values, while the effect of La Niña is opposite (Cai et al., 2018; Gao et al., 2013; Zwart et al., 2016). Continental moisture recycling or local convection during the NSM season has limited impact on $\delta^{18}O_p$ relative to the integrated regional convective activities during the EASM season. As we discussed in section 4.1, however, the difference in $\delta^{18}O_w$ between El Niño and La Niña years is influenced by both the $\delta^{18}O_p$ and the precipitation seasonality. Indeed, there is more summer monsoon precipitation in June-to-September during La Niña years (1988-1989) but more SPR in March-April during El Niño years (1991-1992), though the annual precipitation amounts are similar (Figure 5b). We find that the $\delta^{18}O_w$ variability is broadly consistent with the variation in the ratio of EASM/NSM precipitation amount during 1988-1992 (Figure 5). Unfortunately, the data series of the Changsha GNIP station is too short (5 years) to evaluate the relationship between the EASM/NSM ratio and $\delta^{18}O_p$ in the SPR region. Therefore, we used the average precipitation data from 11 meteorological stations (1951-2014) in the Jiangxi Province and the eastern Hunan Province (Fig. 2, i.e. from the core area of the SPR) and the $\delta^{18}O$ data obtained from the

IsoGSM simulation (1979-2009) to examine the relationship between ENSO, AO, $\delta^{18}O_w$ and precipitation amount in the SPR region on interannual timescales.

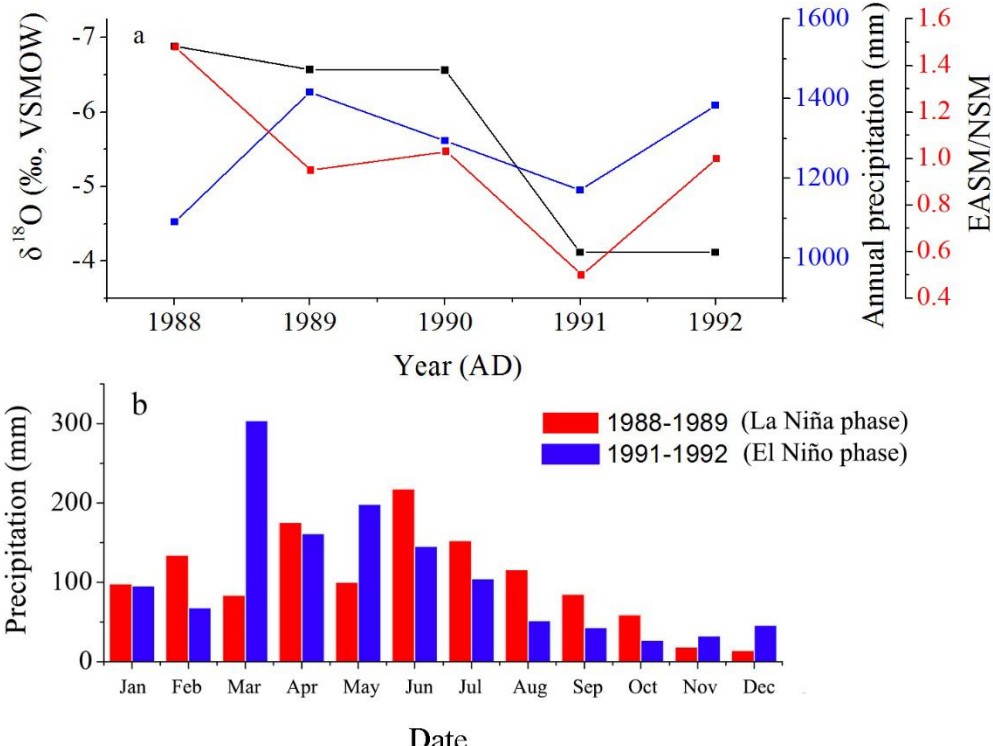

**Figure 5**. Comparison between ENSO events, precipitation amount and $\delta^{18}O_w$ at the Changsha GNIP station for the period 1988-1992. (a) Comparison between annual precipitation amount, $\delta^{18}O_w$, and the EASM/NSM ratio. (b) Comparison of mean monthly precipitation amount between La Niña (1988-1989) and El Niño (1991-1992) years. In this calculation, the temporal coverage of the annual precipitation and the precipitation $\delta^{18}O_w$ is from January to December, the EASM precipitation is from May to September and the NSM precipitation is from January to April and from October to December.

We calculated correlation coefficients between the simulated $\delta^{18}O_w$, the SOI, the MEI, the EASM/NSM ratio, and the annual, EASM, and NSM precipitation amounts for 1979-2009 (Table 2 and Figure 6). The results show that the time series of the simulated $\delta^{18}O_w$ data significantly correlates with the SOI (r=-0.52, *p<0.01*) and the MEI (r=0.51, *p<0.01*), consistent with the positive relationship between the ENSO index and $\delta^{18}O_w$ observed in modern precipitation (Huang et al., 2017; Tan, 2014; Yang et al., 2016) as well as in the $\delta^{18}O$ records of speleothems and tree-ring cellulose (Tan, 2016; Xu et al., 2013, 2016a, 2016b; Zhang et al., 2018). Furthermore, the same relationship holds for the Changsha GNIP station (Figure 5a). This indicates that the precipitation $\delta^{18}O_w$ is higher (lower) during the El Niño (La Niña) phase in the SPR region. There is, however, no significant correlation between the $\delta^{18}O_w$ with the annual, EASM or NSM precipitation amount. This indicates that on interannual timescales, the $\delta^{18}O_w$ is not controlled by the annual or EASM precipitation amount in Southeast China, consistent with the

result based on instrumental data from the Changsha station (Figure 5a) and other studies (Tan et al., 2014; Yang et al., 2016). The time series of the simulated $\delta^{18}O_w$ data correlates with the EASM/NSM ratio (r=-0.36, *p<0.05*) (Figure 6 and Table 2) suggesting that the precipitation $\delta^{18}O_w$ may be influenced by the precipitation seasonality (i. e., EASM/NSM ratio) modulated by ENSO on interannual timescales.

Applying a 2-year smoothing, the time series of the simulated $\delta^{18}O_w$ data significantly correlates with the annual precipitation (r=-0.89, *p<0.01*), the EASM precipitation (r=-0.91, *p<0.01*) and the EASM/NSM ratio (r=-0.81, *p<0.01*) (Table 2 and Figure 6). This indicates that on interannual to decadal timescales the precipitation $\delta^{18}O_w$ might reflect changes in EASM precipitation amount and also the annual precipitation amount and the EASM/NSM ratio, because the EASM/NSM ratio and annual precipitation amount are significantly dominated by the EASM precipitation amount (Table 2).

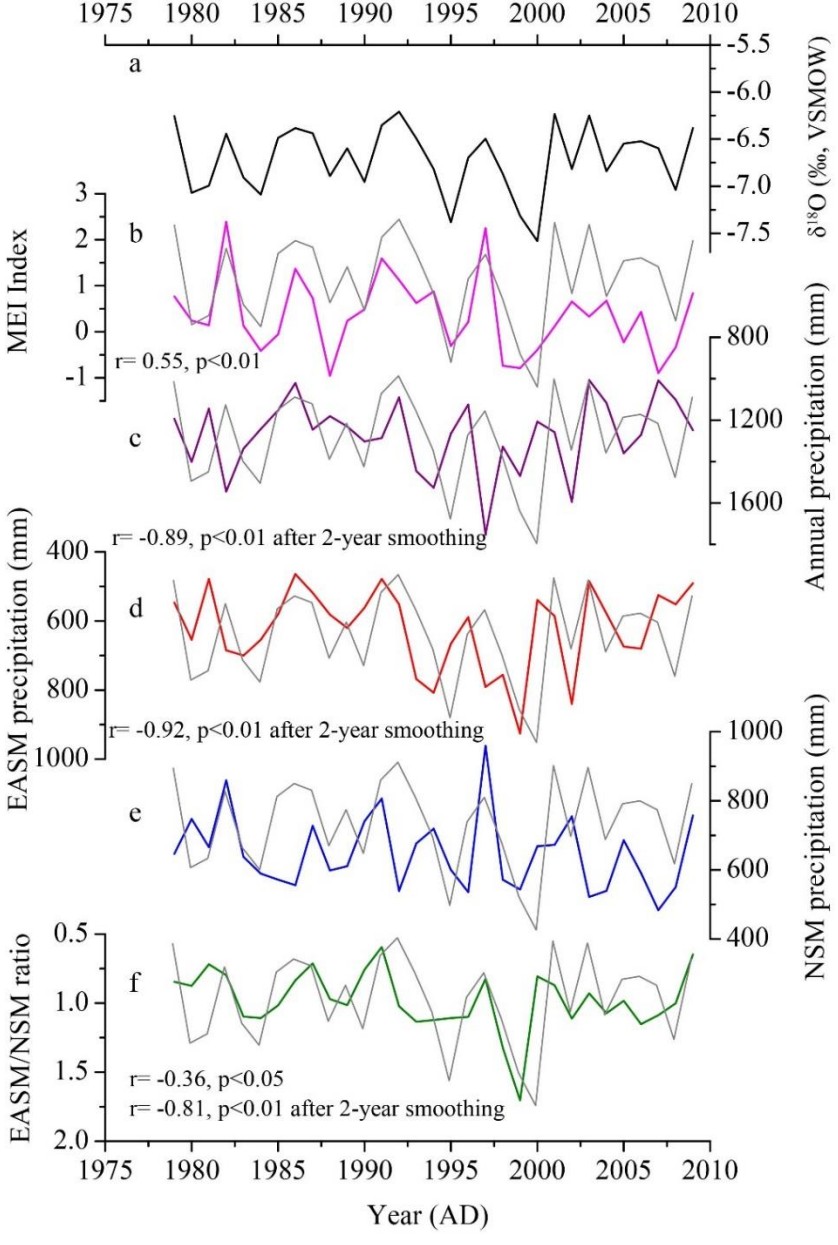

**Figure 6**. Correlation between the time series of the $\delta^{18}O_w$ (a, black line, from May to next April), MEI (b, pink line, from October to next June), annual (c, purple line, from May to next April), EASM (d, red line, from May to September) and NSM (e, blue line, from October to next April) precipitation amount, and the EASM/NSM ratio (f, green line) in the SPR region for 1979-2009. The correlation coefficient between $\delta^{18}O_w$ with MEI index and EASM/NSM ratio is 0.55 (p<0.01) and -0.36 (p<0.05), respectively. Applying a 2-year smoothing $\delta^{18}O_w$ is significantly correlated with annual precipitation (r=-0.89, p<0.01), EASM precipitation (r=-0.92, p<0.01) and the EASM/NSM ratio (-0.81, p<0.01).

**Table 2**. Correlation coefficients between the time series of precipitation $\delta^{18}O_w$, MEI, the EASM/NSM ratio, and the annual, EASM, and NSM precipitation amount in the SPR region for 1979-2009. * indicates significant correlation at the 0.05 level (2-tailed). ** indicates significant correlation at the 0.01 level (2-tailed). The temporal coverage of the annual precipitation and the precipitation $\delta^{18}O_w$ is from May to next April, the EASM precipitation and the NSM precipitation is from May to September and from October to next April. The temporal coverage of the MEI and the SOI is from October to April.

| | SOI | MEI | Annual precipitation | EASM precipitation | NSM precipitation | EASM/NSM ratio |
|---|---|---|---|---|---|---|
| $\delta^{18}O_w$ | -0.52** | 0.55** | -0.12 | -0.31 | 0.13 | -0.36* |
| $\delta^{18}O_w$ (2-year smoothing) | | | -0.89** | -0.92** | -0.28 | -0.81** |
| EASM precipitation (2-year smoothing) | | | 0.92** | | 0.20 | 0.92** |

To explore the relationship between ocean-atmospheric circulation (e.g., ENSO, AO) and the seasonal precipitation amount, we calculated correlation coefficients between the SOI, MEI, AO index, the EASM/NSM ratio, and the annual, EASM, and NSM precipitation amount for 1951-2010 (Table 3). The mean value of October-next June SOI correlates with the EASM precipitation amount (r=0.26, *p<0.05*), NSM precipitation amount (r=-0.51, *p<0.01*) and the EASM/NSM ratio (r=0.52, *p<0.01*) (Table 3). The mean value of October-next June MEI correlates with the EASM precipitation amount (r=-0.29, *p<0.05*), NSM precipitation amount (r=0.54, *p<0.01*) and the EASM/NSM ratio (r=-0.55, *p<0.01*) (Table 3). This indicates, on interannual timescales, decreased EASM precipitation during the developing stage of El Niño and increased NSM precipitation during the mature stage of El Niño, resulting in lower EASM/NSM ratios during El Niño phases and vice versa. There is, however, no significant correlation between the SOI, MEI and the annual precipitation amount. In addition, the

EASM/NSM ratio significantly correlates with the EASM (r=0.64, *p<0.01*) and the NSM (r=-0.70, *p<0.01*) (Table 3).

Previous studies found that decreased summer rainfall in the south of the Yangtze River occurs during the developing stage of El Niño, resulting from a southward shift of the subtropical high associated with colder SST in the western tropical Pacific and weak convective activities in the South China Sea and the Philippines (Huang and Wu, 1989; Zhang et al., 1999). Kong and Tu (2003) found that there is less EASM rainfall in May-September in the lower reaches of the Yangtze River valley during 14 El Niño events since the 1950s. The same relationship is observed in the May-October rainfall reconstruction based on tree-ring cellulose $\delta^{18}O$ (Xu et al., 2016a) during El Niño phases. Cooler summer SST in the western Pacific led to a weakened western Pacific Subtropical High resulting in less rainfall during May-October in the middle to lower reaches of the Yangtze River (Liu and Li, 2011; Xu et al., 2016a), and vice versa. Increased rainfall in autumn, winter and spring (i.e. NSM) occurred in southern China during the mature stage of El Niño (Wan et al., 2008b; Wang et al. 2000; Zhang et al., 1999; Zhang et al., 2015; Zhou, 2011; Zhou and Wu, 2010). During these phases, lower-level southwesterly anomalies over the South China Sea transport more moisture into Southeast China, leading to increased NSM precipitation (Wang et al., 2000; Zhang et al., 1999; Zhou, 2011; Zhou and Wu, 2010). These conclusions are consistent with our findings in the SPR region. It is notable that there is no significant variation in EASM precipitation amount in our study area during the decaying stage of El Niño, although increased summer rainfall was observed in southern China (Huang and Wu, 1989).

We also find that the May AO index significantly and negatively correlates with the annual (r=-0.42, *p<0.01*) and EASM (r=-0.39, *p<0.01*) precipitation amount in the SPR region (Table 3). This is consistent with previous observations that the positive May AO index is followed by decreased summer precipitation amount in the lower Yangtze River valley (Gong and Ho, 2002; He et al., 2017). It was suggested that a stronger May AO is associated with a northwards movement of the summer jet stream, leading to drier conditions in the lower Yangtze River. The positive spring AO gives rise to warmer equatorial SSTs between 150°-180 °E and weakens summer subtropical high in the western North Pacific. Consequently, decreased summer precipitation occurs in the lower Yangtze River (Gong et al., 2011). There is, however, no significant correlation between the AO index and the NSM precipitation amount and the EASM/NSM ratio in the SPR region (Table 3). This might be because the influence of AO on the winter climate varied spatially and temporally, resulting from the unstable relationship between the AO index and the East Asian winter monsoon (He et al., 2017; Li et al., 2014). This indicates that the AO mainly influences the changes in EASM and annual precipitation amount but not the precipitation seasonality (i. e., EASM/NSM ratio) in the SPR region. The February AO index positively correlates with precipitation amount in February (r=0.28, *p<0.05*).

Given the relationship between $\delta^{18}O_w$, SOI, MEI, and seasonal precipitation amount, we find that less EASM during the developing stages of El Niño and more NSM precipitation during the mature stages of El Niño lead to lower EASM/NSM ratios, resulting in higher $\delta^{18}O_w$ values in the SPR region during El Niño phases, and vice versa. We therefore suggest that, over the SPR region the precipitation seasonality (i.e., the EASM/NSM ratio) modulated by ENSO primarily influences the interannual variability of $\delta^{18}O_w$. The AO mainly influences changes in EASM and annual precipitation amount, but not the precipitation seasonality (i. e., EASM/NSM ratio) in the SPR region.

**Table 3**. Correlation coefficients between the time series of the MEI, the EASM/NSM ratio, and the annual, EASM, and NSM precipitation amount in the SPR region for 1951-2010. * indicates significant correlation at the 0.05 level (2-tailed). ** indicates significant correlation at the 0.01 level (2-tailed). The temporal coverage of the annual precipitation and the $\delta^{18}O_w$ is from May to next April, the EASM precipitation is from May to September and the NSM precipitation is from October to next April. The temporal coverage of the SOI and MEI is from October to next June.

|  | Annual precipitation | EASM precipitation | NSM precipitation | EASM/NSM ratio |
|---|---|---|---|---|
| SOI (Oct-next Jun) | -0.15 | 0.26* | -0.51** | 0.52** |
| MEI (Oct-next Jun) | 0.15 | -0.29* | 0.54** | -0.55** |
| AO (May) | -0.42** | -0.39** | -0.17 | 0.18 |
| Annual precipitation |  | 0.67** | 0.69** | -0.05 |
| EASM precipitation |  |  | -0.07 | 0.64** |
| NSM precipitation |  |  |  | -0.70** |

4.3 Implication for paleoclimatic reconstructions

Although speleothem $\delta^{18}O$ records have massively improved our understanding of the EASM variability on different timescales, the significance and quantification of these proxy records is still a subject of debate, because speleothem $\delta^{18}O$ is influenced by several competing factors. We emphasize that the spatial differences in seasonal precipitation over the MRC are key to understand the speleothem $\delta^{18}O$-climate relationship. Figure 1 illustrates that (1) Wanxiang (Zhang et al., 2008), Dayu (Tan et al., 2009), Huangye (Tan et al., 2010), Wuya (Tan et al., 2014), Shihua (Li et al., 2017), and Xiaobailong (Tan et al., 2017) caves are located in the northern and southwestern part of the MRC, where $\delta^{18}O_w$ is primarily controlled by the EASM and annual precipitation amount. Therefore, these records show a

significant correlation with the instrumental precipitation and the regional drought/flood (D/F) index obtained from historical documents (e.g., Li et al., 2017; Liu et al., 2008; Tan et al., 2009, 2010, 2014, 2017; Zhang et al., 2008). (2) Dongge (Yuan et al., 2004), Heshang (Hu et al., 2008), Hulu (Wang et al.,

2001), Yuhua (Jiang et al., 2012) and E'mei (Zhang et al., 2018) caves are located in Southeast China, where $\delta^{18}O_w$ is not only affected by EASM precipitation but also by NSM precipitation. Hence, according to Wang et al. (2001), speleothem $\delta^{18}O$ from Hulu cave reflects the ratio of summer to winter precipitation amount. Factors related to the NSM (e.g., moisture source, integrated regional convection, precipitation seasonality, winter temperature) have also been taken into consideration in the interpretation

of speleothem $\delta^{18}O$ in Southeast China (e.g., Baker et al., 2015; Cai et al., 2018; Cheng et al., 2016; Clemens et al., 2010; Dayem et al., 2010; Zhang et al., 2018). On the other hand, the high percentage of NSM precipitation with relatively high $\delta^{18}O_p$ values in Southeast China should be an important reason why $\delta^{18}O_w$ and speleothem $\delta^{18}O$ are much lower and their variability is much larger in southwestern China than in Southeast China (Li et al., 2016; Liu et al., 2010; Zhang et al., 2018), except for the

influence of integrated regional convection and moisture source and transport distances during the EASM season.

We find that the precipitation seasonality modulated by ENSO mainly controls the $\delta^{18}O_w$ values in the SPR region, with lower (higher) EASM/NSM ratios associated with El Niño (La Niña) phases resulting in higher (lower) $\delta^{18}O_w$ values. Therefore, we suggest that the interannual variability of

490 speleothem $\delta^{18}O$ in the SPR region is primarily controlled by precipitation seasonality (i.e., the EASM/NSM ratio) modulated by ENSO. In addition, the ENSO index in the SPR region also significantly correlates with the EASM precipitation amount on interannual timescales and the precipitation $\delta^{18}O_w$ negatively correlates with the EASM precipitation amount on interannual to decadal timescales, implying that additional studies are needed to disentangle the main driving factor(s) (e.g.,

EASM precipitation amount vs. EASM/NSM ratio) operating on different timescales. Few speleothem $\delta^{18}O$ records have been published for the SPR region so far (Jiang et al., 2012; Zhang et al., 2018). Such long-term records, however, are critically needed to examine the climate-proxy relationship both on interannual and on decadal to millennial timescales.

**5 Conclusions**

We find that the distribution of seasonal precipitation amount in southeastern China, especially in the SPR region, is different from other regions of the MRC for the time interval of this study (1951-2014 AD). In the SPR region, the mean precipitation amount of the EASM is equivalent to that of the NSM. However, in northern and southwestern regions of the MRC, the mean percentage of EASM to the annual precipitation amount exceeds 70%. The seasonal $\delta^{18}O_p$ in the MRC shows broadly consistent variations

with relatively low and high values for EASM and NSM precipitation, respectively. The low $\delta^{18}O_p$ values

associated with winter precipitation in northern region of the MRC, however, represent only a minor contribution to $\delta^{18}O_w$. Thus, the NSM precipitation in the SPR region also has an important effect on $\delta^{18}O_w$, but the $\delta^{18}O_w$ in northern and southwestern regions is primarily influenced by EASM precipitation.

Based on a statistical analysis of the ENSO index, simulated $\delta^{18}O$ data and seasonal precipitation amount in the SPR region, we find that less (more) EASM and more (less) NSM precipitation leads to a lower (higher) EASM/NSM ratio resulting in higher (lower) $\delta^{18}O_w$ in the SPR region during El Niño (La Niña) phases. The AO mainly influences the changes in EASM and annual precipitation amount but not the precipitation seasonality (e.g., EASM/NSM ratio) in the SPR region. Recognizing this spatial difference in seasonal precipitation is essential for a robust interpretation of speleothem $\delta^{18}O$ in the MRC. On interannual timescales, speleothem $\delta^{18}O$ variability in northern and southwestern regions of the MRC is primarily influenced by the EASM or the annual precipitation amount. In the SPR region, however, precipitation seasonality (i.e., the EASM/NSM ratio) modulated by ENSO plays a key role in governing speleothem $\delta^{18}O$ variability, although integrated regional convection and moisture source and transport distance may also have subordinate impacts.

**6 Author Contributions**

H.W.Z designed the research and wrote the first draft of the manuscript. H.C. Y.J.C. C.S. and A.S. helped to revise the manuscript. G.K and H.Y.L helped to get the IsoGSM simulation data. All authors discussed the results and provided input on the manuscript.

**7 Competing interests**

The authors declare no competing interests.

**Acknowledgments**

Thanks to Ming Tan for reviewing the manuscript and giving many constructive suggestions. Thanks to Kristina Krklec, Hui Tang and Zhongyin Cai for helping with the back-trajectory analyses and moisture source calculations. This study was supported by the NSFC (41502166), the China Postdoctoral Science Foundation (2015M580832), the State Key Laboratory of Loess and Quaternary Geology (SKLLQG1046) and the Key Laboratory of Karst Dynamics, Ministry of Land and Resources of the People's Republic of China (MLR) and GZAR (KDL201502).

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
