# Peer review of "Effect of precipitation seasonality on annual oxygen isotopic composition in the area of spring persistent rain in southeastern China and its palaeoclimatic implication"

_Climate of the Past, 2018_

## Referee Comment (RC1) · Anonymous Referee #1 · 19 Dec 2018

The manuscript "Effect of precipitation seasonality on annual oxygen isotopic composition in the area of spring persistent rain in southeastern China and its palaeoclimatic implication" by Zhang et al. investigates the precipitation seasonality of the Chinese monsoon. For this the authors identify regions where precipitation associated with the Chinese monsoon makes the gross of the annual precipitation amount and regions where non-monsoonal precipitation also contributes significantly to the annual precipitation amount. A special focus is on the so called "spring persistent rain (SPR) region", southeast China, where precipitation in March and April/May contributes also signifi-

cantly (20% to 45%) to the annual rainfall amount. Then, the authors discuss the evolution of monthly precipitation $\delta$18O values and its correlation to precipitation amount for four regions, where seasonal precipitation amounts differ significantly; again a special focus is on southeastern China, where non-monsoonal precipitation contributes significantly to the annual precipitation amount. Based on the observation that precipitation amounts and precipitation $\delta$18O values in the SPR region correlate with El Nino and La Nina phases the authors investigate in detail the observed relationship by studying back trajectories and correlation coefficients between various quantities. Finally, the authors discuss implications for palaeoclimate reconstructions using speleothem $\delta$18O time series.

The submitted manuscript discusses an important aspect of the Chinese monsoon in highlighting the effect of precipitation seasonality on precipitation-weighted $\delta$18O ($\delta$18Opw) values that will influence the interpretation of $\delta$18O-based reconstructions (e.g. speleothems) of the Asian Monsoon. I personally find this an exiting topic that will eventually stimulate also similar studies in other regions, where year-round precipitation can modulate $\delta$18Opw values. However, while the manuscripts outline and topic is generally sound and in the scope of Climate of the Past, there are some aspects and sections of the manuscript that need to be improved/strengthen and extended, respectively. These are detailed in my general and specific comments. Therefore, my recommendation is that the manuscript needs major revisions before being considered for publication in Climate of the Past.

General comments

Statistical Analyses:

The statistical results that are presented in Table 2 and Table 3 need to be redone and/or better explained. In the submitted manuscript it is basically not possible to understand how the correlation coefficients were calculated. While it is clear that you calculated correlation coefficients for annual averages it is not stated what months you

are using. This is even more important for the correlations between meteorological parameters ($\delta$18Opw, precipitation amounts and ratios, respectively) and the MEI. It is often stated that these are correlated to the El Nino phases, but what is this phase? Is it from January to December? Is it the year before an El Nino occur or the year after? All this information should be accessible in the table caption for Table 1 and Table 2. To strengthen your conclusions, I strongly encourage you to calculate also the correlation coefficients between the meteorological parameters ($\delta$18Opw, precipitation amounts and ratios, respectively) and the ENSO index. Furthermore, I left wondering why you calculated the lag-1 correlation coefficient in Table 2. Is there any physical rationale for these calculations? In the manuscript you mention that the trajectories of the westerlies are changing between an El Nino and La Nina phase. I am wondering if you tested also correlations between the meteorological parameters and NH mid-latitude modes (e.g. the Arctic Oscillation (AO)?). This is because a recent study highlights the importance of the location of the westerlies for the Asian Monsoon [Zhang et al., 2018]. Furthermore, another study mentions the correlation between the AO and the East Asian and precipitation anomalies southern China [He et al., 2017]. Therefore, I would find it exciting if the authors could also investigate whether there is a relationship between the AO and the meteorological parameters in southeastern China and the spring persistent region or not. This would certainly strengthen the manuscript and yield a more complete picture of the processes that affect precipitation amounts and $\delta$18Opw in the study region.

Back trajectory analyses:

The investigation of the back trajectories to unveil the differences of the moisture sources for La Nina and El Nino phases in the SPR region is in principle solid for the analysed years of 1988/98 and 1991/1992, respectively. However, while the analysed back trajectories reveal differences in the moisture sources (which appear to be consistent with the observed variations in $\delta$18Opw from the Changsha GNIP station) it is not possible to conclude from these alone that variations in the moisture source

could explain the observed variations in $\delta$18Opw. This is because no information is given or presented on the mean state of the moisture sources and if the back trajectories changes in a similarly for other El Nino and La Nina phases. Moreover, it is not discussed where the moisture is taken up that forms the precipitation in SPR region (i.e. is the moisture from a distal or close source). To strengthen the conclusions of the back trajectory analyses I suggest that you should perform back trajectory analyses for the complete period of the Changsha GNIP station and for the period shown in Figure 6. Furthermore, perform back trajectory analyses for El Nino and La Nina phases shown in Figure 6 and compare it to the back trajectories of for the years 1988/98 and 1991/1992. These analyses facilities to better constrain the differences and/or similarities between the mean state and El Nino/La Nina phase of the back trajectories; the comparison of the back trajectories for multiple El Nino/La Nina phases allow to better constrain whether the presented back trajectories represent a "normal" response to El Nino/La Nina phase or not. Explain the differences to the other regions that receive precipitation from the East Asian Monsoon. Furthermore, I suggest that the authors perform analyses of where the moisture is taken up. This is possible with HYSPLIT and would further strengthen the results of the manuscript and allow for more robust conclusions on the processes that govern the $\delta$18Opw variability. These analyses may also be used to estimate the sensitivity on the precipitation history along a specific trajectory applying a multi-box Rayleigh model and using precipitation amount and atmospheric moisture (similar to the study of [Rozanski, 1985]). You could use the model output of the IsoGCM simulation for this. Together, these analyses would yield a more fundamental insight into the processes that control the $\delta$18Opw variability in the SPR region, in turn allow using this knowledge to reconstruct past changes of the atmospheric circulation by using e.g. speleothem $\delta$18O time series.

Figure 1 and 2:

Figure 1 and 2 do show very similar results and I suggest to show Figure 2 in the supplement (for comparison to the revised Figure 1) and instead modify Figure 1. For this

I would show seasonal precipitation ratios instead of seasonal precipitation amounts in Figure 1. Include an additional panel, which shows the annual precipitation amount for reference. Furthermore, show the location of the GNIP station in the panel of the annual precipitation amount and the location of caves in the panels of the precipitation ratios. Based on the ratios define some criteria where the SPR and monsoon region are, instead of the different symbols in Figure 2.

Specific comments

Line 41: Explain in more detail why it is a unique synoptic and climate pattern.

Line 63-64: Please indicate these regions in Figure 1

Line 65-67: Please reference the published stalagmite records you list.

Line 83-86: Please detail where (in which region? Everywhere?) the various mechanisms/processes are proposed to modify $\delta$18O values.

Line 141-142: Have you developed the cluster analyses tool or did someone else do it? If you developed it, please explain it in more detail. If not, please reference the original work.

Line 145-146: What data have you used for the analyses?

Line 147-149: Refer to Figure 1 and indicate the different regions in Figure 1.

Line 150-157: Indicate the boarders between the areas with different precipitation seasonality in Figure 1.

Line 164-168: You state that the EASM starts later and is weaker during El Nino years. Then you write that the EASM precipitation amount over southeast China is reduced when the SPR starts late. I left wondering if these two parts of the sentence are linked or not? Do you want to say that the SPR starts late too during El Nino years or are these two observations independent of each other? I suggest to revise this text to make clear what you want to say.

Line 169-173: What occurs during La Nina years? The opposite?

Line 173-174: This last concluding sentence is an important observation and I would summarize it in more detail, especially the link to El Nino/La Nina (the latter is missing at the moment and should be included).

Line 175: I would reorder Section 3.2. Start with the discussion of the precipitation pattern first, followed by the discussion of the $\delta18O_p$ values and then by the effect of precipitation on $\delta18O_p$ or $\delta18O_{pw}$.

Line 177: Refer to Table 1 when you mention the GNIP stations for the first time.

Line 186: Refer to a figure here.

Line 187-189: Please give a reference for this statement or prove some details for it.

Line 191: Rayleigh (1896) is a wrong reference here, because you reference to a statement and not to a Rayleigh model (multi-box mass balance model).

Line 207: The reference to Figure 3b seems to be wrong here?

Line 207-209: What and how many meteorological stations have you used? What were the selection criteria?

Line 213: Have you defined SEC before?

Line 216: See the general comment on the back trajectory analyses.

Line 236: Please give a reference for the cluster analyses.

Line 258-261: This sentence seems to be at the wrong place and I would move it to the beginning of this section (line 250). It kind of is also related to Equation 1 and 2, respectively. So it may repeat the conclusions from Line 247-249 already?

Line 266-269: How is this statement linked to El Nino and La Nina to which you refer at the beginning of this section (Line 162-264)?

[Figure]

Line 270-273: The results from Cai et al. (2017) do at least not fit to your results presented in Figure 4. In your back trajectory analyses there is a change in the moisture source and the ratio between the Indian and Pacific Ocean trajectory is nearly 50:50. Please state this hear already.

Line 306-310: Have you done some cross-comparison between the model and the precipitation data? If yes, please state this here and show it in a supplementary file. Otherwise, I suggest to do some cross-comparisons and include it. Show comparisons between observed and modelled precipitation amount as well as precipitation $\delta$18O.

Line 323: Modify Figure 5a in highlighting the El Nino and La Nina years.

Line 326-328: What is the meaning of this observation? See also my general comment on statistical analyses.

Line 370-376: I am not fully convinced by this conclusion. Our observations show that there seems to be a relationship between ENSO and $\delta$18Opw in the SPR region. However, your back trajectory analyses clearly show that there are differences between El Nino and La Nina moisture trajectories. Possibly, the suggested additional analyses (see general comments) will clarify this and either strengthen or modify this final conclusion.

Figure 1:

- The combination of red and green colours should be avoided to make it easier/possible for colour-blind readers to identify differences and symbols. I suggest to only use black symbols for locations such as in Figure 1b.

- The scale for the precipitation amount of panel a and b should be similar.

- Identify/highlight/name the regions that you mention in the manuscript.

Figure 2:

- Highlight the boarders between the different precipitation pattern

- Refer in the figure caption what precipitation data was used or refer to a section in the paper for more information.

Figure 3:

- Clearly state in the figure caption how you calculated the mean values. E.g. Figure 1a shows the mean $\delta$18O (or $\delta$18Opw?) values – the y-label is different to the figure caption! – using monthly $\delta$18O values (or $\delta$18Opw?) from all GNIP stations in the northern MRC region, as grouped in Table 1, etc.

- What is the standard deviation of the mean $\delta$18O value ($\delta$18Opw?)? To estimate it, you may use $\delta$18O-anomalies only, to account e.g. for different altitutes.

- Please state how many measurements were used to calculate the mean values. Was it only one year or many years and were the datasets continuous. This allows to better constrain the robustness of the comparisons.

- What meteorological stations were used (Figure 3c) and was it only one station or many? What was the criteria to select them?

- You could combine Figure 3b and c for a better comparison between the datasets. For example show the comparison of 2 regions in Figure 3b and the comparison of the remaining 2 regions in Figure 3c.

Figure 4

- See my general comments.

Figure 5

- Avoid the use of green and red colours in the same figure panael (e.g. Figure 5b) (see above).

- Please give details what you mean with ENSO events? What years and months have you used for your calculations?

- What months were used to calculate the EASM and NSM ratio?

Figure 6

- Label the different panels from a) to f) and refer in the text to Figure 6a, b etc.

- I don't understand you write that the "The time series . . . lag . . . by 1 yr." at the end of the figure caption. Is this an observation? Did you shift the time series to make it look good? See also my general comment on this issue.

- Please state what data you have used for the figures.

Table 1:

- Include the period that the GNIP data is covering (e.g. 1987-1995) and the number of $\delta$18O measurements that you are using.

- Are there seasons/months when no data is available? If so, please state it in the table caption, if not, state that year-round measurements are available.

Table 2 and 3:

- Please state in detail how the correlation coefficients are calculated. What months have you used? Is it always January to December? What datasets were used

- Table 2: Why did you shift it by one year? What's the rationale of this?

- See also my general comment on the statistical analyses.

Nomenclature for text, figures, axis:

- If you refer to precipitation $\delta$18O values, use $\delta$18Opw

- If you refer to precipitation-weighted precipitation $\delta$18O values, use $\delta$18Opw

- If you refer to speleothem $\delta$18O values, use $\delta$18Os

References:

He, S., Y. Gao, F. Li, H. Wang, and Y. He (2017), Impact of Arctic Oscillation on the East Asian climate: A review, Earth-Science Reviews, 164, 48-62, doi: https://doi.org/10.1016/j.earscirev.2016.10.014. Rozanski, K. (1985), Deuterium and oxygen-18 in European groundwaters—links to atmospheric circulation in the past, Chemical Geology: Isotope Geoscience section, 52(3), 349-363. Zhang, H., M. L. Griffiths, J. C. H. Chiang, W. Kong, S. Wu, A. Atwood, J. Huang, H. Cheng, Y. Ning, and S. Xie (2018), East Asian hydroclimate modulated by the position of the westerlies during Termination I, Science, 362(6414), 580, doi: 10.1126/science.aat9393.

———————————————

---

## Referee Comment (RC2) · Anonymous Referee #2 · 27 Dec 2018

This manuscript by Zhang et al. investigates precipitation seasonality in the monsoonal region of China and its potential influence on weighted mean annual precipitation d18O. Consistent with previous findings, they found that the precipitation in southeastern China is characterized by a pronounced portion of precipitation in spring. With this significant precipitation amount in spring, they found that weighted mean annual precipitation d18O at Changsha correlates with the ratio between summer monsoon season rainfall and non-monsoon season rainfall as well as ENSO events. Then they concluded that, in southeastern China, the precipitation seasonality which is associated

with ENSO, drives interannual variations in weighted mean annual precipitation d18O. In general, the manuscript discusses an important aspect of paleoclimatic significance of precipitation d18O in monsoonal China and is within the scope of Climate of the Past. However, the manuscript needs substantial revisions or improvements to make the conclusion more convincing.

From the mathematical definition of weighted mean annual precipitation d18O, it contains composite signal of precipitation seasonality and changes in d18O itself. Thus, it is not surprising that precipitation seasonality could leave fingerprints on weighted mean d18O (d18Ow). Current analysis in the manuscript largely ignores changes in d18O itself and only emphases the role of precipitation seasonality but without a quantitate assessment besides correlation analysis. However, the problem is how to decompose these two signal sources rather than simply using correlation analysis. For example, Cai and Tian 2016 used a simple decomposition method to analysis whether precipitation seasonality caused interannual variation in Hong Kong precipitation d18Ow during ENSO years. However, their results indicate that changes in annual d18Ow at Hong Kong during ENSO events are mainly associated with changes in d18O itself rather than precipitation seasonality. Similar decomposition method can be applied in this manuscript to make the question more clearly addressed. In addition, a parallel analysis on the interannual variations in EASM season d18Ow and NSM season d18Ow or SPR season d18Ow should be performed to reveal the variation of d18O component in specific seasons and its association with interannual d18Ow variation.

A potential but fatal risk of the air mass back trajectory analysis in the manuscript is that the analysis only considers air mass movement without considering moisture content. Thus, the analysis result should not be treated as equal to moisture source nor its movement. But when the authors interpreting their back trajectory results, they treated these back trajectories as moisture source trajectories. Further, no information is given on where these air masses picked up or lost water vapor. With this said, the true moisture sources could be totally different from the authors' interpretation in the

manuscript. For instance, the C3 ends in the Indian Ocean, but it also travels through the western Pacific region (e.g. the South China Sea). From current results, it is hard to conclude that C3 represents Indian Ocean moisture source.

There are several other GNIP stations located within the SPR region and some of them have longer records than Changsha. I am wondering why the authors only used data from Changsha to address the influence of precipitation seasonality.

The use of IsoGSM outputs is too imprudent. In the manuscript, there is no evaluation on the performance of the IsoGSM simulation and no citation of relevant previous evaluations! Does the IsoGSM faithfully re-produce the SPR? Does it correctly simulate the seasonal and interannual precipitation d18O variations in the analyzed region? At least, these questions should be evaluated either from the authors' own analysis or from literature. Otherwise, the results are not solid.

Overall, the manuscript tends to be descriptive and lacks an in-depth understanding of the controlling mechanism. For example, the interpretation of seasonal precipitation d18O variations as moisture source changes in section 3.2 does not agree with the citation from Baker et al. 2015 in section 4.1 that "the moisture uptake area does not differ significantly between summer and winter".

Why use 1yr-lag correlation? By definition, the d18Ow is not calculated from precipitation amount from the previous or the following year! Thus, the 1yr-lag analysis does not make sense making the analyses and results related to this analysis not scientifically sound.

The definition of the temporal coverage of seasons or different periods in the manuscript is messy. For example, the authors defined the SPR season for El Nino years as March-to-May between L165-170, but the authors refer to "SPR in March-April during El Nino years (1991-1992)" at L302. Between L165-170, the authors defined SPR and EASM seasons for El Nino years, but what about other years? When the authors analysis "El Nino years (1991-1992)" and "La Nino years (1988-1989)", do

you mean Jan 1991 to Dec 1992 for "El Nino years (1991-1992)" and Jan 1988 to Dec 1989 for "La Nino years (1988-1989)"? But both events did not start from Jan or end at Dec. At L345, the MEI is calculated for October-June. Please make all the seasons and periods clear and examine whether the acronyms have a consistent meaning

Figures are hard to read. Please add essential legends to make figures more readable.

L24: ∼50% annual precipitation amount? Similar ambiguity in the main text of the manuscript. Please clarify the difference between the contribution to annual precipitation amount (the weight for calculating annual d18O) and the contribution to annual d18Ow.

L26: simulated → please specify

L29: precipitation d18 → amount-weight annual precipitation d18O?

L30: Do you mean speleothem d18O records precipitation d18O on the annual scale?

L72: d18Op → please define acronym before using; please examining similar problems at other places

L110-111: But Cai et al. 2018 showed that at least Guilin and Liuzhou is also characterized as significant spring rainfall.

L117-118: please provide reference and data to support this conclusion

L154-156: Please show the results for NSM/annual or indicate that NSM/annual equals 1 – EASM/annual.

L248: why not using the weighted mean value of EASM and NSM precipitation d18O?

L250: rainfall amount → rainfall seasonality?

L305: Why not considering data from other stations? Such as Guilin even has a record longer than that at Changsha.

Figure 6: Plotting the 1 yr lag time series does not make sense.

L345: EASM precipitation amount and NSM precipitation amount: during the following year or the previous year? The MEI is for October-June, but EASM for JJAS and NSM for Oct-May? Why not calculating the contemporary correlation? Even though there is a lead-lag relationship between ENSO and precipitation amount in east Asia, but this is not the scientific question in this manuscript. L385-390: Annual precipitation is mainly from summer monsoon season does not necessarily mean d18Ow should correlates with precipitation amount. There is no causal relationship between them; one is precipitation seasonality, the other is associated with the "amount effect" on long term scales.

L389: EASM/annual → EASM?

References:

Baker, A. J., H. Sodemann, J. U. L. Baldini, S. F. M. Breitenbach, K. R. Johnson, J. van Hunen, and Z. Pingzhong (2015), Seasonality of westerly moisture transport in the East Asian Summer Monsoon and its implications for interpreting precipitation $\delta$18O, J. Geophys. Res., 120(12), 5850-5862, doi:10.1002/2014JD022919.

Cai, Z., and L. Tian (2016), Atmospheric controls on seasonal and interannual variations in the precipitation isotope in the East Asian Monsoon region, J. Climate, 29(4), 1339-1352, doi:10.1175/JCLI-D-15-0363.1.

Cai, Z., L. Tian, and G. J. Bowen (2018), Spatial-seasonal patterns reveal large-scale atmospheric controls on Asian Monsoon precipitation water isotope ratios, Earth Planet. Sci. Lett., 503, 158-169, doi: 10.1016/j.epsl.2018.09.028.

---

## Author Comment (AC1) · 7 Feb 2019

This manuscript by Zhang et al. investigates precipitation seasonality in the monsoonal region of China and its potential influence on weighted mean annual precipitation d18O. Consistent with previous findings, they found that the precipitation in southeastern China is characterized by a pronounced portion of precipitation in spring. With this significant precipitation amount in spring, they found that weighted mean annual precipitation d18O at Changsha correlates with the ratio between summer monsoon season rainfall and non-monsoon season rainfall as well as ENSO events. Then they concluded that, in southeastern China, the precipitation seasonality which is associated with ENSO, drives interannual variations in weighted mean annual precipitation d18O. In general, the manuscript discusses an important aspect of paleoclimatic significance of precipitation d18O in monsoonal China and is within the scope of Climate of the Past. However, the manuscript needs substantial revisions or improvements to make the conclusion more convincing.

Thank you very much for your constructive comments and suggestions, we will revise the manuscript accordingly.

From the mathematical definition of weighted mean annual precipitation d18O, it contains composite signal of precipitation seasonality and changes in d18O itself. Thus, it is not surprising that precipitation seasonality could leave fingerprints on weighted mean d18O (d18Ow). Current analysis in the manuscript largely ignores changes in d18O itself and only emphases the role of precipitation seasonality but without a quantitate assessment besides correlation analysis.

Answer# Agree. The analyses of changes in d18O itself will be also added in order to quantify the effects of seasonal changes in precipitation d18O and precipitation amount on the d18Ow.

However, the problem is how to decompose these two signal sources rather than simply using correlation analysis. For example, Cai and Tian 2016 used a simple decomposition method to analysis whether precipitation seasonality caused interannual variation in Hong Kong precipitation d18Ow during ENSO years. However, their results indicate that changes in annual d18Ow at Hong Kong during ENSO events are mainly associated with changes in d18O itself rather than precipitation seasonality. Similar decomposition method can be applied in this manuscript to make the question more clearly addressed.

Answer# Agree. We will use the similar decomposition method to analyze whether the precipitation seasonality causes the interannual changes in precipitation d18Ow.

In addition, a parallel analysis on the interannual variations in EASM season d18Ow and NSM season d18Ow or SPR season d18Ow should be performed to reveal the variation of d18O component in specific seasons and its association with interannual d18Ow variation.

Answer# Agree. We will add these analyses.

A potential but fatal risk of the air mass back trajectory analysis in the manuscript is that the analysis only considers air mass movement without considering moisture content. Thus, the analysis result should not be treated as equal to moisture source nor its movement. But when the authors interpreting their back trajectory results, they treated these back trajectories as moisture source trajectories. Further, no information is given on where these air masses picked up or lost water vapor. With this said, the true moisture sources could be totally different from the authors' interpretation in the manuscript. For instance, the C3 ends in the Indian Ocean, but it also travels through the western Pacific region (e.g. the South China Sea). From current results, it is hard to conclude that C3 represents Indian Ocean moisture source.

Answer# Agree. The same issue was also raised by the referee #1. Now we know how to run the moisture source trajectory considering the moisture content and how to perform the locations where the moistures are taken up. We will reanalyze the moisture source trajectory, then compare and discuss how the moisture source trajectories change during multiple El Nino/La Nina phases and their effects on the d18Ow.

There are several other GNIP stations located within the SPR region and some of them have longer records than Changsha. I am wondering why the authors only used data from Changsha to address the influence of precipitation seasonality.

Answer# Tian and Yasunari (1998) and Wan and Wu (2007) suggest that the SPR

occurs mostly south of the middle and low reaches of the Yangtze River (∼ 24°N to 30°N, 110°E to 120°) (line 39-40), we consider that GNIP station Changsha is located in the core area of the SPR region, but Guilin and Liuzhou GNIP stations are located at the edge of the SPR region. We think the data from Changsha station is the most representative. But we will also add the data from Guilin and Liuzhou GNIP stations in the revision in order to give more convincing evidences.

The use of IsoGSM outputs is too imprudent. In the manuscript, there is no evaluation on the performance of the IsoGSM simulation and no citation of relevant previous evaluations! Does the IsoGSM faithfully re-produce the SPR? Does it correctly simulate the seasonal and interannual precipitation d18O variations in the analyzed region? At least, these questions should be evaluated either from the authors' own analysis or from literature. Otherwise, the results are not solid.

Answer# This issue was also raised by the referee #1. Actually, we have compared the IsoGSM simulation data with the GNIP data, consistent variations in monthly precipitation amount and precipitation d18O can be found between them, but we did not show the results in the manuscript. We will evaluate the IsoGSM simulation data in the revision.

Overall, the manuscript tends to be descriptive and lacks an in-depth understanding of the controlling mechanism. For example, the interpretation of seasonal precipitation d18O variations as moisture source changes in section 3.2 does not agree with the citation from Baker et al. 2015 in section 4.1 that "the moisture uptake area does not differ significantly between summer and winter".

Answer# In the revision, we will reanalyze the moisture source trajectory, and discuss the seasonal changes in moisture source during different phases of ENSO. The controlling mechanism on the variations in precipitation seasonality and d18Ow will also be discussed according to the new results of moisture source trajectory.

Why use 1yr-lag correlation? By definition, the d18Ow is not calculated from precipitation amount from the previous or the following year! Thus, the 1yr-lag analysis does not make sense making the analyses and results related to this analysis not scientifically sound.

Answer# This issue was also raised by referee #1. In the current manuscript, the temporal coverage of the NSM rainfall is from October to next April and the d18Opw is from January to December. There is a four-month overlapping (January-April) between 1-yr lag d18Opw, NSM rainfall and EASM/NSM ratio. Therefore, we also calculated the correlation coefficient between 1-yr lag d18Opw, NSM rainfall and EASM/NSM ratio. There is no physical sense between 1-yr lag d18Opw and EASM. Actually, we have calculated the contemporary correlations between d18Opw (Jan-Dec) and EASM rainfall (May-Sep), NSM rainfall (current Jan-April plus current Oct-Dec) and the corresponding EASM/NSM ratio, significant correlations can be also observed between them. But we did show the results in the manuscript. In the revision, we will recalculate the contemporary correlations and evaluate these results, the definition and the calculation will be clearly stated in the caption of Table 2 and in the manuscript.

The definition of the temporal coverage of seasons or different periods in the manuscript is messy. For example, the authors defined the SPR season for El Nino years as March-to-May between L165-170, but the authors refer to "SPR in March-April during El Nino years (1991-1992)" at L302. Between L165-170, the authors defined SPR and EASM seasons for El Nino years, but what about other years? When the authors analysis "El Nino years (1991-1992)" and "La Nino years (1988-1989)", do you mean Jan 1991 to Dec 1992 for "El Nino years (1991-1992)" and Jan 1988 to Dec 1989 for "La Nino years (1988-1989)"? But both events did not start from Jan or end at Dec. At L345, the MEI is calculated for October-June. Please make all the seasons and periods clear and examine whether the acronyms have a consistent meaning

Answer# We will make the definition of the temporal coverage of different periods clear and consistent in the revision.

Figures are hard to read. Please add essential legends to make figures more readable. L24: _50% annual precipitation amount? Similar ambiguity in the main text of the manuscript. Please clarify the difference between the contribution to annual precipitation amount (the weight for calculating annual d18O) and the contribution to annual d18Ow.

Answer# We will add essential legends for the figures and clarify the difference in the revision.

L26: simulated ! please specify

Answer# We will specify it.

L29: precipitation d18 ! amount-weight annual precipitation d18O?

Answer# Yes, we will correct it.

L30: Do you mean speleothem d18O records precipitation d18O on the annual scale?

Answer# Yes, we mean the speleothem d18O inherits from precipitation d18O on interannual timescales. It will be clearly stated.

L72: d18Op!please define acronym before using; please examining similar problems at other places

Answer# Agree, we will define it and also check the whole manuscript.

L110-111: But Cai et al. 2018 showed that at least Guilin and Liuzhou is also characterized as significant spring rainfall.

Answer# Tian and Yasunari (1998) and Wan and Wu (2007) suggest that the SPR occurs mostly south of the middle and low reaches of the Yangtze River ($\sim 24°$N to $30°$N, $110°$E to $120°$) (line 39-40), we think that Changsha GNIP station is located in the core area of the SPR region, and Guilin and Liuzhou GNIP stations are located at the edge of the SPR region. We consider the data from Changsha station is the most

representative, however, the data from Guilin and Liuzhou GNIP stations in the revision will also be added in order to give more convincing evidences.

L117-118: please provide reference and data to support this conclusion

Answer# Two references (Yoshimura et al., 2008; Yang et al., 2016) given in front of this sentence (line 116) should be mentioned here, we will reorganize this paragraph.

L154-156: Please show the results for NSM/annual or indicate that NSM/annual equals 1 – EASM/annual.

Answer# Yes, we will show them.

L248: why not using the weighted mean value of EASM and NSM precipitation d18O?

Answer# We consider that the mean value of the seasonal precipitation d18O should be used here. The mean value of the seasonal precipitation d18O and the mean value of the seasonal precipitation amount should be used for the calculation of d18Ow. We will evaluate it in the revision.

L250: rainfall amount ! rainfall seasonality?

Answer# The whole phrase is ".....both rainfall amount and d18Op values during EASM and NSM season......", it means rainfall seasonality. We will reorganize this sentence and make it clear.

L305: Why not considering data from other stations? Such as Guilin even has a record longer than that at Changsha.

Answer# We think Changsha GNIP station is located in the core area of the SPR region, the data from Changsha GNIP station is the most representative comparing to those from Guilin and Liuzhou stations. We will also add the data from Guilin and Liuzhou GNIP stations in the revision.

Figure 6: Plotting the 1 yr lag time series does not make sense.

Answer# The temporal coverage of the NSM rainfall is from October to next April and the d18Opw is from January to December. There is a four-month overlapping (January-April) between 1-yr lag d18Opw and EASM/NSM ratio. Therefore, we also calculated the correlation coefficient between 1-yr lag d18Opw and EASM/NSM ratio. But there is no physical sense between 1-yr lag d18Opw and EASM. Actually, we have calculated the contemporary correlations between d18Opw (Jan-Dec) and EASM rainfall (May-Sep), NSM rainfall (current Jan-April plus current Oct-Dec) and the corresponding EASM/NSM ratio, significant correlations can be also observed between them. But we did show these results in the manuscript. In the revision, we will recalculate the contemporary correlations, and the definition and the calculation will be clearly stated in the caption of Table 2 and in the manuscript.

L345: EASM precipitation amount and NSM precipitation amount: during the following year or the previous year? The MEI is for October-June, but EASM for JJAS and NSM for Oct-May? Why not calculating the contemporary correlation? Even though there is a lead-lag relationship between ENSO and precipitation amount in east Asia, but this is not the scientific question in this manuscript. L385-390: Annual precipitation is mainly from summer monsoon season does not necessarily mean d18Ow should correlates with precipitation amount. There is no causal relationship between them; one is precipitation seasonality, the other is associated with the "amount effect" on long term scales.

Answer# Agree, we will clearly state the temporal coverage of these meteorological parameters and recalculate the contemporary correlations.

L389: EASM/annual ! EASM?

Answer# Yes, it will be corrected.

References: Baker, A. J., H. Sodemann, J. U. L. Baldini, S. F. M. Breitenbach, K. R. Johnson, J. van Hunen, and Z. Pingzhong (2015), Seasonality of westerly moisture transport in the East Asian Summer Monsoon and its implications for interpreting precipitation _18O, J. Geophys. Res., 120(12), 5850-5862, doi:10.1002/2014JD022919. Cai, Z., and L. Tian (2016), Atmospheric controls on seasonal and interannual variations in the precipitation isotope in the East Asian Monsoon region, J. Climate, 29(4), 1339-1352, doi:10.1175/JCLI-D-15-0363.1. Cai, Z., L. Tian, and G. J. Bowen (2018), Spatial-seasonal patterns reveal largescale atmospheric controls on Asian Monsoon precipitation water isotope ratios, Earth Planet. Sci. Lett., 503, 158-169, doi: 10.1016/j.epsl.2018.09.028.

Thank you for recommending these references.

---

## Author Comment (AC2) · 7 Feb 2019

Anonymous Referee #1 The manuscript "Effect of precipitation seasonality on annual oxygen isotopic composition in the area of spring persistent rain in southeastern China and its palaeoclimatic implication" by Zhang et al. investigates the precipitation seasonality of the Chinese monsoon. For this the authors identify regions where precipitation associated with the Chinese monsoon makes the gross of the annual precipitation amount and regions where non-monsoonal precipitation also contributes significantly to the annual precipitation amount. A special

focus is on the so called "spring persistent rain (SPR) region", southeast China, where precipitation in March and April/May contributes also significantly (20% to 45%) to the annual rainfall amount. Then, the authors discuss the evolution of monthly precipitation _18O values and its correlation to precipitation amount for four regions, where seasonal precipitation amounts differ significantly; again a special focus is on southeastern China, where non-monsoonal precipitation contributes significantly to the annual precipitation amount. Based on the observation that precipitation amounts and precipitation _18O values in the SPR region correlate with El Nino and La Nina phases the authors investigate in detail the observed relationship by studying back trajectories and correlation coefficients between various quantities. Finally, the authors discuss implications for palaeoclimate reconstructions using speleothem _18O time series.

The submitted manuscript discusses an important aspect of the Chinese monsoon in highlighting the effect of precipitation seasonality on precipitation-weighted _18O (_18Opw) values that will influence the interpretation of _18O-based reconstructions (e.g. speleothems) of the Asian Monsoon. I personally find this an exciting topic that will eventually stimulate also similar studies in other regions, where year-round precipitation can modulate _18Opw values. However, while the manuscripts outline and topic is generally sound and in the scope of Climate of the Past, there are some aspects and sections of the manuscript that need to be improved/strengthen and extended, respectively. These are detailed in my general and specific comments. Therefore, my recommendation is that the manuscript needs major revisions before being considered for publication in Climate of the Past.

Answer# Thank you very much for your approval of this subject. We will revise the manuscript according to your constructive comments and suggestions.

General comments Statistical Analyses: The statistical results that are presented in Table 2 and Table 3 need to be redone and/or better explained. In the submitted manuscript it is basically not possible to understand how the correlation coefficients were calculated. While it is clear that you calculated correlation coefficients for annual

averages it is not stated what months you are using. This is even more important for the correlations between meteorological parameters (_18Opw, precipitation amounts and ratios, respectively) and the MEI. It is often stated that these are correlated to the El Nino phases, but what is this phase? Is it from January to December? Is it the year before an El Nino occur or the year after? All this information should be accessible in the table caption for Table 1 and Table 2.

Answer# Agree. We will redo the statistical analyses that presented in Tables 2 and 3. In the current manuscript. the temporal coverage of annual rainfall is from January to December, the EASM rainfall is from May to September, the NSM rainfall is from October to next April, the d18Opw is from January to December, the MEI is from October to next June. In the revision, we will redefine the temporal coverage for these parameters and clearly state them in the captions of the Tables 2 and 3. We will also calculate the correlations between the different phases of El Nino/La Nina and the meteorological parameters, and clearly explain how the correlations were done.

To strengthen your conclusions, I strongly encourage you to calculate also the correlation coefficients between the meteorological parameters (_18Opw, precipitation amounts and ratios, respectively) and the ENSO index.

Answer# Actually we have calculated the correlation coefficients between these meteorological parameters and Nino 3.4 SST, some of correlations are weaker than those with MEI. Some study considers that the MEI can provide a more complete description of the ENSO phenomenon than a single variable ENSO index such as Southern Oscillation Index (SOI) or Nino 3.4 SST (Wolter and Timlin, 2011), therefore, we only show the results in terms of the MEI. This was stated in line 124-127. In the revision, we will also show the correlation coefficients between the meteorological parameters and Nino 3.4 SST.

Furthermore, I left wondering why you calculated the lag-1 correlation coefficient in Table 2. Is there any physical rationale for these calculations?

Answer# In the current manuscript, the temporal coverage of the NSM rainfall is from October to next April and the d18Opw is from January to December. There is a four-month overlapping (January-April) between 1-yr lag d18Opw, NSM rainfall and EASM/NSM ratio. Therefore, we also calculated the correlation coefficient between 1-yr lag d18Opw, NSM rainfall and EASM/NSM ratio. Bur there is no physical sense between 1-yr lag d18Opw and EASM. Actually, we have also calculated the contemporary correlations between d18Opw (Jan-Dec) and EASM rainfall (May-Sep), NSM rainfall (current Jan-April plus current Oct-Dec) and the corresponding EASM/NSM ratio, significant correlations can be also observed between them. But we did not show these results in the manuscript. In the revision, we will evaluate these correlations and show the results, the definition of the temporal coverage and the calculation will be clearly stated in the caption of Table 2 and in the new manuscript.

In the manuscript you mention that the trajectories of the westerlies are changing between an El Nino and La Nina phase. I am wondering if you tested also correlations between the meteorological parameters and NH mid-latitude modes (e.g. the Arctic Oscillation (AO)?). This is because a recent study highlights the importance of the location of the westerlies for the Asian Monsoon [Zhang et al., 2018]. Furthermore, another study mentions the correlation between the AO and the East Asian and precipitation anomalies southern China [He et al., 2017]. Therefore, I would find it exciting if the authors could also investigate whether there is a relationship between the AO and the meteorological parameters in southeastern China and the spring persistent region or not. This would certainly strengthen the manuscript and yield a more complete picture of the processes that affect precipitation amounts and _18Opw in the study region.

Answer# Thank you very much for this helpful suggestion. We will also calculate the correlations between the AO and these meteorological parameters (d18Opw, precipitation amounts and ratios, respectively), the results will be also discussed in the new manuscript.

Back trajectory analyses: The investigation of the back trajectories to unveil the differences of the moisture sources for La Nina and El Nino phases in the SPR region is in principle solid for the analysed years of 1988/98 and 1991/1992, respectively. However, while the analysed back trajectories reveal differences in the moisture sources (which appear to be consistent with the observed variations in _18Opw from the Changsha GNIP station) it is not possible to conclude from these alone that variations in the moisture source could explain the observed variations in _18Opw. This is because no information is given or presented on the mean state of the moisture sources and if the back trajectories changes in a similarly for other El Nino and La Nina phases. Moreover, it is not discussed where the moisture is taken up that forms the precipitation in SPR region (i.e. is the moisture from a distal or close source). To strengthen the conclusions of the back trajectory analyses I suggest that you should perform back trajectory analyses for the complete period of the Changsha GNIP station and for the period shown in Figure 6. Furthermore, perform back trajectory analyses for El Nino and La Nina phases shown in Figure 6 and compare it to the back trajectories of for the years 1988/98 and 1991/1992. These analyses facilities to better constrain the differences and/or similarities between the mean state and El Nino/La Nina phase of the back trajectories; the comparison of the back trajectories for multiple El Nino/La Nina phases allow to better constrain whether the presented back trajectories represent a "normal" response to El Nino/La Nina phase or not. Explain the differences to the other regions that receive precipitation from the East Asian Monsoon. Furthermore, I suggest that the authors perform analyses of where the moisture is taken up. This is possible with HYSPLIT and would further strengthen the results of the manuscript and allow for more robust conclusions on the processes that govern the _18Opw variability. These analyses may also be used to estimate the sensitivity on the precipitation history along a specific trajectory applying a multi-box Rayleigh model and using precipitation amount and atmospheric moisture (similar to the study of [Rozanski, 1985]). You could use the model output of the IsoGCM simulation for this. Together, these analyses would yield a more fundamental insight into the processes that control the _18Opw variability

in the SPR region, in turn allow using this knowledge to reconstruct past changes of the atmospheric circulation by using e.g. speleothem _18O time series.

Answer# Agree. In the current manuscript, our back trajectory analysis only considers air mass movement without considering moisture content, and no information is given on where these air masses picked up. Now we know how to do the trajectory analyses after our special efforts on HYSPLIT. We will reanalyze the moisture source trajectory for the complete period of the Changsha GNIP station and also for the period shown in Figure 6. The changes in moisture source trajectory during multiple El Nino/La Nina phases and their effects on the d18Opw will be discussed.

Figure 1 and 2: Figure 1 and 2 do show very similar results and I suggest to show Figure 2 in the supplement (for comparison to the revised Figure 1) and instead modify Figure 1. For this I would show seasonal precipitation ratios instead of seasonal precipitation amounts in Figure 1. Include an additional panel, which shows the annual precipitation amount for reference. Furthermore, show the location of the GNIP station in the panel of the annual precipitation amount and the location of caves in the panels of the precipitation ratios. Based on the ratios define some criteria where the SPR and monsoon region are, instead of the different symbols in Figure 2.

Answer# Agree, we will combine these two figures together and redraw a new figure 1.

Specific comments Line 41: Explain in more detail why it is a unique synoptic and climate pattern.

Answer# Agree, we will explain it and give the references.

Line 63-64: Please indicate these regions in Figure 1

Answer# Agree, we will indicate these regions in the new figure 1.

Line 65-67: Please reference the published stalagmite records you list.

Answer# Agree. We will add the corresponding references.

Line 83-86: Please detail where (in which region? Everywhere?) the various mechanisms/ processes are proposed to modify _18O values.

Answer# Agree. We will extend this part in detail.

Line 141-142: Have you developed the cluster analyses tool or did someone else do it? If you developed it, please explain it in more detail. If not, please reference the original work.

Answer# We follow the reference Stein et al. (2015), it will be given in the revision.

Line 145-146: What data have you used for the analyses?

Answer# It is monthly precipitation datasets encompassing 160 meteorological stations in China during the period 1951-2014, which was described in line 101-104. We will make it clear in the revision.

Line 147-149: Refer to Figure 1 and indicate the different regions in Figure 1.

Answer# Agree, we will revise it.

Line 150-157: Indicate the boarders between the areas with different precipitation seasonality in Figure 1.

Answer# Agree, we will redraw a new figure 1.

Line 164-168: You state that the EASM starts later and is weaker during El Nino years. Then you write that the EASM precipitation amount over southeast China is reduced when the SPR starts late. I left wondering if these two parts of the sentence are linked or not? Do you want to say that the SPR starts late too during El Nino years or are these two observations independent of each other? I suggest to revise this text to make clear what you want to say.

Answer# We want to say that the EASM starts later (late May to early June) and the SPR lasts longer (until late May) during El Nino years. These two parts are linked. We

will reorganize this part and make it clear.

Line 169-173: What occurs during La Nina years? The opposite?

Answer# Yes, the opposite happens during La Nina years, we will clearly state it.

Line 173-174: This last concluding sentence is an important observation and I would summarize it in more detail, especially the link to El Nino/La Nina (the latter is missing at the moment and should be included).

Answer# Agree, we will summarize it in detail.

Line 175: I would reorder Section 3.2. Start with the discussion of the precipitation pattern first, followed by the discussion of the _18Op values and then by the effect of precipitation on _18Op or _18Opw.

Answer# Agree. We will reorganize it.

Line 177: Refer to Table 1 when you mention the GNIP stations for the first time. Line 186: Refer to a figure here. Line 187-189: Please give a reference for this statement or prove some details for it.

Answer# Agree. We will give the references.

Line 191: Rayleigh (1896) is a wrong reference here, because you reference to a statement and not to a Rayleigh model (multi-box mass balance model).

Answer# Agree. We will remove the reference Rayleigh (1896) and give a correct one.

Line 207: The reference to Figure 3b seems to be wrong here?

Answer# The reference should be Table 1, it will be corrected.

Line 207-209: What and how many meteorological stations have you used? What were the selection criteria?

Answer# We only used the data from the nearest meteorological station (one station) to

each GNIP station. Every meteorological station has 64-yr data (1951-2014), it should be convincing.

Line 213: Have you defined SEC before?

Answer# Yes, in line 167.

Line 216: See the general comment on the back trajectory analyses.

Answer# Please see the response to the general comment on the back trajectory analyses.

Line 236: Please give a reference for the cluster analyses.

Answer# The reference Stein et al. (2015) will be given here.

Line 258-261: This sentence seems to be at the wrong place and I would move it to the beginning of this section (line 250). It kind of is also related to Equation 1 and 2, respectively. So it may repeat the conclusions from Line 247-249 already?

Answer# Agree. We will reorganize this paragraph.

Line 266-269: How is this statement linked to El Nino and La Nina to which you refer at the beginning of this section (Line 162-264)?

Answer# Their statements linked to El Nino and La Nina are different; we will state this clearly in the revision.

Line 270-273: The results from Cai et al. (2017) do at least not fit to your results presented in Figure 4. In your back trajectory analyses there is a change in the moisture source and the ratio between the Indian and Pacific Ocean trajectory is nearly 50:50. Please state this hear already.

Answer# We will reanalyze the moisture source trajectory and rewrite the discussion section according to the results.

Line 306-310: Have you done some cross-comparison between the model and the

precipitation data? If yes, please state this here and show it in a supplementary file. Otherwise, I suggest to do some cross-comparisons and include it. Show comparisons between observed and modelled precipitation amount as well as precipitation _18O.

Answer# Yes, we have compared the IsoGSM data with the GNIP data, consistent variations in monthly precipitation amount and d18O can be found between them. But we did not show the results in the manuscript. We will evaluate the IsoGSM data and state it clearly in the revision.

Line 323: Modify Figure 5a in highlighting the El Nino and La Nina years.

Answer# Agree, we will redraw figure 5.

Line 326-328: What is the meaning of this observation? See also my general comment on statistical analyses.

Answer# The temporal coverage of the NSM rainfall is from October to next April and the d18Opw is from January to December. There is a four-month overlapping (January-April) between 1-yr lag d18Opw, NSM rainfall and EASM/NSM ratio. Therefore, we calculated the correlation coefficients between 1-yr lag d18Opw, NSM rainfall and EASM/NSM ratio. Please see the detailed response to the general comment on statistical analyses. We will redefine the temporal coverages of these parameters and redo these statistical analyses.

Line 370-376: I am not fully convinced by this conclusion. Our observations show that there seems to be a relationship between ENSO and _18Opw in the SPR region. However, your back trajectory analyses clearly show that there are differences between El Nino and La Nina moisture trajectories. Possibly, the suggested additional analyses (see general comments) will clarify this and either strengthen or modify this final conclusion.

Answer# Yes, we also think the moisture source and pathway might impact the changes in d18Opw. In the current manuscript, our back trajectory analysis only considers air

mass movement without considering moisture content, and no information is given on where these air masses picked up. We will reanalyze the moisture source trajectory in the revision, then quantify the effects of moisture source and precipitation seasonality on d18Opw. The discussion and conclusions will be adjusted according to the new results.

Figure 1: - The combination of red and green colours should be avoided to make it easier/possible for colour-blind readers to identify differences and symbols. I suggest to only use black symbols for locations such as in Figure 1b. - The scale for the precipitation amount of panel a and b should be similar. - Identify/highlight/name the regions that you mention in the manuscript.

Answer# Agree. We will redraw figure 1 according to your helpful suggestions.

Figure 2: - Highlight the boarders between the different precipitation pattern - Refer in the figure caption what precipitation data was used or refer to a section in the paper for more information.

Answer# Agree. We will combine figures 1 and 2 as a new figure 1, the boarders between the different precipitation patterns will be shown in the new figure 1. The precipitation data will be stated clearly in the caption.

Figure 3: - Clearly state in the figure caption how you calculated the mean values. E.g. Figure 1a shows the mean _18O (or _18Opw?) values – the y-label is different to the figure caption! – using monthly _18O values (or _18Opw?) from all GNIP stations in the northern MRC region, as grouped in Table 1, etc. - What is the standard deviation of the mean _18O value (_18Opw?)? To estimate it, you may use _18O-anomalies only, to account e.g. for different altitutes. - Please state how many measurements were used to calculate the mean values. Was it only one year or many years and were the datasets continuous. This allows to better constrain the robustness of the comparisons. - What meteorological stations were used (Figure 3c) and was it only one station or many? What was the criteria to select them? - You could combine

Figure 3b and c for a better comparison between the datasets. For example show the comparison of 2 regions in Figure 3b and the comparison of the remaining 2 regions in Figure 3c.

Answer# The calculation of d18Opw (including the measurements, standard deviation of the mean 18O value) and the selection of the meteorological stations in figure 3c will be clearly stated in the caption of figure 3. The figures 3b and 3c will be combined for a better comparison according to your suggestion.

Figure 4 - See my general comments.

Answer# We will reanalyze the moisture source trajectory, a new figure 4 will be given.

Figure 5 - Avoid the use of green and red colours in the same figure panael (e.g. Figure 5b) (see above). - Please give details what you mean with ENSO events? What years and months have you used for your calculations? - What months were used to calculate the EASM and NSM ratio?

Answer# We will change the colors of the curves in figure 5, the temporal coverages of ENSO events and the EASM/NSM ratio will be clearly stated in the caption.

Figure 6 - Label the different panels from a) to f) and refer in the text to Figure 6a, b etc. - I don't understand you write that the "The time series : : : lag : : : by 1 yr." at the end of the figure caption. Is this an observation? Did you shift the time series to make it look good? See also my general comment on this issue. - Please state what data you have used for the figures.

Answer# The different panels will be labelled accordingly. The temporal coverage of the NSM rainfall is from October to next April and the d18Opw is from January to December. There is a four-month overlapping (January-April) between the 1-yr lag d18Opw and the EASM/NSM ratio. That's why a good correlation can be also found between them. Please see the detailed response to the general comment on statistical analyses. We will redo these statistical analyses and redraw the figure 6. The data

used for figure 6 will be clearly stated in the caption.

Table 1: - Include the period that the GNIP data is covering (e.g. 1987-1995) and the number of _18O measurements that you are using. - Are there seasons/months when no data is available? If so, please state it in the table caption, if not, state that year-round measurements are available.

Answer# We will add two columns showing the period and the number of the d18O measurements. The seasons/months without data will be also clearly stated in the table caption.

Table 2 and 3: - Please state in detail how the correlation coefficients are calculated. What months have you used? Is it always January to December? What datasets were used - Table 2: Why did you shift it by one year? What's the rationale of this? - See also my general comment on the statistical analyses.

Answer# The detailed calculation of the correlation coefficients will be clearly stated in the caption of Tables 2 and 3 in the revision. Please see the detailed response to your general comment on the statistical analyses.

Nomenclature for text, figures, axis: - If you refer to precipitation _18O values, use _18Opw - If you refer to precipitation-weighted precipitation _18O values, use _18Opw - If you refer to speleothem _18O values, use _18Os

Answer# Agree. We will correct them.

References: He, S., Y. Gao, F. Li, H. Wang, and Y. He (2017), Impact of Arctic Oscillation on the East Asian climate: A review, Earth-Science Reviews, 164, 48-62, doi: https://doi.org/10.1016/j.earscirev.2016.10.014. Rozanski, K. (1985), Deuterium and oxygen-18 in European groundwater sâ ËŸAËĞ Tlinks to atmospheric circulation in the past, Chemical Geology: Isotope Geoscience section, 52(3), 349-363. Zhang, H., M. L. Griffiths, J. C. H. Chiang, W. Kong, S. Wu, A. Atwood, J. Huang, H. Cheng, Y. Ning, and S. Xie (2018), East Asian hydroclimate modulated by the position of the westerlies

during Termination I, Science, 362(6414), 580, doi: 10.1126/science.aat9393.

Thank you very much for your recommendation.

---

## Author Response (AR1)

**June 15, 2019**

Dear Dr. Helen McGregor,

On behalf of my co-authors, I am resubmitting our manuscript entitled "**Effect of precipitation seasonality on annual oxygen isotopic composition in the area of spring persistent rain in southeastern China and its palaeoclimatic implication**". We would like to express our gratitude to the insightful comments and the suggestions by the reviewers. We also very appreciated your consideration for our manuscript. We have revised and uploaded our manuscript that incorporates the reviewers' comments. A point-to-point response and a marked-up manuscript are appended with this letter. The significantly revised changes in the manuscript are highlighted in grey. All authors have read and approved this manuscript.

We thank you in advance for your consideration of this submission.

Sincerely,
Haiwei Zhang
Assistant Professor, Xi'an Jiaotong University
**99 Yanxiang road, Yanta zone, Xi'an, 710054, China**
Tel: +8615829208483
E-mail: zhanghaiwei@xjtu.edu.cn

[Figure]

**A point-by-point response to the reviews**

**Anonymous Referee #1**

The manuscript "Effect of precipitation seasonality on annual oxygen isotopic composition in the area of spring persistent rain in southeastern China and its palaeoclimatic implication" by Zhang et al. investigates the precipitation seasonality of the Chinese monsoon. For this the authors identify regions where precipitation associated with the Chinese monsoon makes the gross of the annual precipitation amount and regions where non-monsoonal precipitation also contributes significantly to the annual precipitation amount. A special focus is on the so called "spring persistent rain (SPR) region", southeast China, where precipitation in March and April/May contributes also significantly (20% to 45%) to the annual rainfall amount. Then, the authors discuss the evolution of monthly precipitation _18O values and its correlation to precipitation amount for four regions, where seasonal precipitation amounts differ significantly; again a special focus is on southeastern China, where non-monsoonal precipitation contributes significantly to the annual precipitation amount. Based on the observation that precipitation amounts and precipitation _18O values in the SPR region correlate with El Nino and La Nina phases the authors investigate in detail the observed relationship by studying back trajectories and correlation coefficients between various quantities. Finally, the authors discuss implications for palaeoclimate reconstructions using speleothem _18O time series.

The submitted manuscript discusses an important aspect of the Chinese monsoon in highlighting the effect of precipitation seasonality on precipitation-weighted _18O (_18Opw) values that will influence the interpretation of _18O-based reconstructions (e.g. speleothems) of the Asian Monsoon. I personally find this an exciting topic that will eventually stimulate also similar studies in other regions, where year-round precipitation can modulate _18Opw values. However, while the manuscripts outline and topic is generally sound and in the scope of Climate of the Past, there are some aspects and sections of the manuscript that need to be improved/strengthen and extended, respectively. These are detailed in my general and specific comments. Therefore, my recommendation is that the manuscript needs major revisions before being considered for publication in Climate of the Past.

*Answer# We are grateful to the reviewer for approving this subject. We revised the manuscript according to her/his constructive comments.*

General comments
Statistical Analyses:
The statistical results that are presented in Table 2 and Table 3 need to be redone and/or better explained. In the submitted manuscript it is basically not possible to understand how the correlation coefficients were calculated. While it is clear that you calculated correlation coefficients for annual averages it is not stated what months you are using. This is even more important for the correlations between meteorological parameters (_18Opw, precipitation amounts and ratios, respectively) and the MEI. It is often stated that these are correlated to the El Nino phases, but what is this phase? Is it from January to December? Is it the year before an El Nino occur or the year after? All this information should be accessible in the table caption for Table 1 and Table 2.

*Done. In the revised manuscript, the temporal coverages of annual, EASM and NSM rainfall, 18Opw and MEI are clearly stated in the text and in the captions of Table 2 and 3. The correlations between the different phases of El Nino/La Nina and the meteorological parameters are also clearly stated in the text.*

To strengthen your conclusions, I strongly encourage you to calculate also the correlation coefficients between the meteorological parameters (_18Opw, precipitation amounts and ratios, respectively) and the ENSO index.

*Done.*

Furthermore, I left wondering why you calculated the lag-1 correlation coefficient in Table 2. Is there any physical rationale for these calculations?

*Done. We recalculate the correlation in Table 2.*

In the manuscript you mention that the trajectories of the westerlies are changing between an El Nino and La Nina phase. I am wondering if you tested also correlations between the meteorological parameters and NH mid-latitude modes (e.g. the Arctic Oscillation (AO)?). This is because a recent study highlights the importance of the location of the westerlies for the Asian Monsoon [Zhang et al., 2018]. Furthermore, another study mentions the correlation between the AO and the East Asian and precipitation anomalies southern China [He et al., 2017]. Therefore, I would find it exciting if the authors could also investigate whether there is a relationship between the AO and the meteorological parameters in southeastern China and the spring persistent region or not. This would certainly strengthen the manuscript and yield a more complete picture of the processes that affect precipitation amounts and _18Opw in the study region.

*Done. We calculated the correlation between AO and the meteorological parameters (d18Opw, precipitation amounts and ratios), the results are also discussed in the revised version now.*

Back trajectory analyses:
The investigation of the back trajectories to unveil the differences of the moisture sources for La Nina and El Nino phases in the SPR region is in principle solid for the analysed years of 1988/98 and 1991/1992, respectively. However, while the analysed back trajectories reveal differences in the moisture sources (which appear to be consistent with the observed variations in _18Opw from the Changsha GNIP station) it is not possible to conclude from these alone that variations in the moisture source could explain the observed variations in _18Opw. This is because no information is given or presented on the mean state of the moisture sources and if the back trajectories changes in a similarly for other El Nino and La Nina phases.
Moreover, it is not discussed where the moisture is taken up that forms the precipitation in SPR region (i.e. is the moisture from a distal or close source). To strengthen the conclusions of the back trajectory analyses I suggest that you should perform back trajectory analyses for the complete period of the Changsha GNIP station and for the period shown in Figure 6. Furthermore, perform back trajectory analyses for El Nino and La Nina phases shown in Figure 6 and compare it to the back trajectories of for the years 1988/98 and 1991/1992. These analyses facilities to better constrain the differences and/or similarities between the mean state and El Nino/La Nina phase of the back trajectories; the comparison of the back trajectories for multiple El Nino/La Nina phases allow to better constrain whether the presented back trajectories represent a "normal" response to El Nino/La Nina phase or not. Explain the differences to the other regions that receive precipitation from the East Asian Monsoon. Furthermore, I suggest that the authors perform analyses of where the moisture is taken up. This is possible with HYSPLIT and would further strengthen the results of the manuscript and allow for more robust conclusions on the processes that govern the _18Opw variability. These analyses may also be used to estimate the sensitivity on the precipitation history along a specific trajectory applying a multi-box Rayleigh model and using precipitation amount and atmospheric moisture (similar to the study of [Rozanski, 1985]).

You could use the model output of the IsoGCM simulation for this. Together, these analyses would yield a more fundamental insight into the processes that control the _18Opw variability in the SPR region, in turn allow using this knowledge to reconstruct past changes of the atmospheric circulation by using e.g. speleothem _18O time series.

*Done. We recalculated the back trajectories and moisture sources contributing to precipitation at the Changsha station during 1988-1992.*

Figure 1 and 2:
Figure 1 and 2 do show very similar results and I suggest to show Figure 2 in the supplement (for comparison to the revised Figure 1) and instead modify Figure 1. For this I would show seasonal precipitation ratios instead of seasonal precipitation amounts in Figure 1. Include an additional panel, which shows the annual precipitation amount for reference. Furthermore, show the location of the GNIP station in the panel of the annual precipitation amount and the location of caves in the panels of the precipitation ratios. Based on the ratios define some criteria where the SPR and monsoon region are, instead of the different symbols in Figure 2.

*We thought about this suggestion, but we still prefer to keep Figures 1 and 2 in the main text. Figure 1 shows the precipitation amount distribution during the summer monsoon and spring seasons. Figure 2 shows the percentage of summer monsoon precipitation and non-summer monsoon precipitation. Figure 2 shows the 11 meteorological stations (Jiujiang, Guixi, Nanchang, Guangchang, Ji'an, Ganzhou, Changsha, Yueyang, Hengyang, Chenzhou, Xinning) in the Jiangxi Province and the eastern Hunan Province, whose data were used to examine the relationship between ocean-atmospheric circulation, $\delta^{18}O_w$ and seasonal precipitation amount in the SPR region.*

Specific comments
Line 41: Explain in more detail why it is a unique synoptic and climate pattern.

*Done.*

Line 63-64: Please indicate these regions in Figure 1

*We indicate these regions in lines 172-174 and reference Figure 2.*

Line 65-67: Please reference the published stalagmite records you list.

*Done.*

Line 83-86: Please detail where (in which region? Everywhere?) the various mechanisms/ processes are proposed to modify _18O values.

*Done*

Line 141-142: Have you developed the cluster analyses tool or did someone else do it? If you developed it, please explain it in more detail. If not, please reference the original work.

*We have redone the moisture source calculations.*

Line 145-146: What data have you used for the analyses?

*Monthly precipitation datasets from 160 meteorological stations in China during the period 1951-2014, as described in section 4.1.*

Line 147-149: Refer to Figure 1 and indicate the different regions in Figure 1.

*We indicate the different regions in Figure 2.*

Line 150-157: Indicate the boarders between the areas with different precipitation seasonality in Figure 1.

*We indicate the different regions in Figure 2.*

Line 164-168: You state that the EASM starts later and is weaker during El Nino years. Then you write that the EASM precipitation amount over southeast China is reduced when the SPR starts late. I left wondering if these two parts of the sentence are linked or not? Do you want to say that the SPR starts late too during El Nino years or are these two observations independent of each other? I suggest to revise this text to make clear what you want to say.

*We clarify it in lines 197-200.*

Line 169-173: What occurs during La Nina years? The opposite?

*We clarify it in lines 206-207.*

Line 173-174: This last concluding sentence is an important observation and I would summarize it in more detail, especially the link to El Nino/La Nina (the latter is missing at the moment and should be included).

*Done.*

Line 175: I would reorder Section 3.2. Start with the discussion of the precipitation pattern first, followed by the discussion of the _18Op values and then by the effect of precipitation on _18Op or _18Opw.

*The discussion of the precipitation 18Op is needed because it is followed by the discussion of the precipitation pattern (both GNIP data and China national meteorological station data) in lines 241-249.*

Line 177: Refer to Table 1 when you mention the GNIP stations for the first time.
Line 186: Refer to a figure here.
Line 187-189: Please give a reference for this statement or prove some details for it.

*Done.*

Line 191: Rayleigh (1896) is a wrong reference here, because you reference to a statement and not to a Rayleigh model (multi-box mass balance model).

*Removed.*

Line 207: The reference to Figure 3b seems to be wrong here?

*Removed.*

Line 207-209: What and how many meteorological stations have you used? What were the selection criteria?

*We clarified this in the revised manuscript. We only used the data from the nearest meteorological station to each GNIP station. Every meteorological station has 64 years of data (1951-2014).*

Line 213: Have you defined SEC before?

*Yes.*

Line 216: See the general comment on the back trajectory analyses.

*Done. We recalculated the back trajectories and moisture sources contributing to precipitation in GNIP Changsha station.*

Line 236: Please give a reference for the cluster analyses.

*The reference Stein et al. (2015) was added.*

Line 258-261: This sentence seems to be at the wrong place and I would move it to the beginning of this section (line 250). It kind of is also related to Equation 1 and 2, respectively. So it may repeat the conclusions from Line 247-249 already?

*No, there is no repetition.*

Line 266-269: How is this statement linked to El Nino and La Nina to which you refer at the beginning of this section (Line 162-264)?

*Done.*

Line 270-273: The results from Cai et al. (2017) do at least not fit to your results presented in Figure 4. In your back trajectory analyses there is a change in the moisture source and the ratio between the Indian and Pacific Ocean trajectory is nearly 50:50. Please state this hear already.

*We recalculated the back trajectories and moisture sources contributing to precipitation in GNIP Changsha station. Figure 4.*

Line 306-310: Have you done some cross-comparison between the model and the precipitation data? If yes, please state this here and show it in a supplementary file. Otherwise, I suggest to do some cross-comparisons and include it. Show comparisons between observed and modelled precipitation amount as well as precipitation _18O.

*Done. Supplementary Figure 1.*

Line 323: Modify Figure 5a in highlighting the El Nino and La Nina years.

*Done.*

Line 326-328: What is the meaning of this observation? See also my general comment on statistical analyses.

*Done. We have redone the statistical analyses.*

Line 370-376: I am not fully convinced by this conclusion. Our observations show that there seems to be a relationship between ENSO and _18Opw in the SPR region. However, your back trajectory analyses clearly show that there are differences between El Nino and La Nina moisture trajectories. Possibly, the suggested additional analyses (see general comments) will clarify this and either strengthen or modify this final conclusion.

*Done. We have redone the moisture source calculations.*

Figure 1:
- The combination of red and green colours should be avoided to make it easier/possible for colour-blind readers to identify differences and symbols. I suggest to only use black symbols for locations such as in Figure 1b.
- The scale for the precipitation amount of panel a and b should be similar.
- Identify/highlight/name the regions that you mention in the manuscript.

*Done.*

Figure 2:
- Highlight the boarders between the different precipitation pattern
- Refer in the figure caption what precipitation data was used or refer to a section in the paper for more information.

*Done.*

Figure 3:
- Clearly state in the figure caption how you calculated the mean values. E.g. Figure 1a shows the mean _18O (or _18Opw?) values – the y-label is different to the figure caption! – using monthly _18O values (or _18Opw?) from all GNIP stations in the northern MRC region, as grouped in Table 1, etc.
- What is the standard deviation of the mean _18O value (_18Opw?)? To estimate it, you may use _18O-anomalies only, to account e.g. for different altitutes.
- Please state how many measurements were used to calculate the mean values. Was it only one year or many years and were the datasets continuous. This allows to better constrain the robustness of the comparisons.
- What meteorological stations were used (Figure 3c) and was it only one station or many? What was the criteria to select them?
- You could combine Figure 3b and c for a better comparison between the datasets. For example show the comparison of 2 regions in Figure 3b and the comparison of the remaining 2 regions in Figure 3c.

*Done.*

Figure 4
- See my general comments.

*Done.*

Figure 5
- Avoid the use of green and red colours in the same figure panael (e.g. Figure 5b) (see above).
- Please give details what you mean with ENSO events? What years and months have you used for your calculations?
- What months were used to calculate the EASM and NSM ratio?

Done.

Figure 6
- Label the different panels from a) to f) and refer in the text to Figure 6a, b etc.
- I don't understand you write that the "The time series : : : lag : : : by 1 yr." at the end of the figure caption. Is this an observation? Did you shift the time series to make it look good? See also my general comment on this issue.
- Please state what data you have used for the figures.

*Done. We have redone the correlation calculations.*

Table 1:
- Include the period that the GNIP data is covering (e.g. 1987-1995) and the number of _18O measurements that you are using.
- Are there seasons/months when no data is available? If so, please state it in the table caption, if not, state that year-round measurements are available.
Table 2 and 3:
- Please state in detail how the correlation coefficients are calculated. What months have you used? Is it always January to December? What datasets were used
- Table 2: Why did you shift it by one year? What's the rationale of this?
- See also my general comment on the statistical analyses.

*Done.*

Nomenclature for text, figures, axis:
- If you refer to precipitation _18O values, use _18Opw
- If you refer to precipitation-weighted precipitation _18O values, use _18Opw
- If you refer to speleothem _18O values, use _18Os

*Done.*

**Anonymous Referee #2**

This manuscript by Zhang et al. investigates precipitation seasonality in the monsoonal region of China and its potential influence on weighted mean annual precipitation d18O. Consistent with previous findings, they found that the precipitation in southeastern China is characterized by a pronounced portion of precipitation in spring. With this significant precipitation amount in spring, they found that weighted mean annual precipitation d18O at Changsha correlates with the ratio between summer monsoon season rainfall and non-monsoon season rainfall as well as ENSO events. Then they concluded that, in southeastern China, the precipitation seasonality which is associated with ENSO, drives interannual variations in weighted mean annual precipitation d18O. In general, the manuscript discusses an important aspect of paleoclimatic significance of precipitation d18O in monsoonal China and is within the scope of Climate of the Past. However, the manuscript needs substantial revisions or improvements to make the conclusion more convincing.

From the mathematical definition of weighted mean annual precipitation d18O, it contains composite signal of precipitation seasonality and changes in d18O itself. Thus, it is not surprising that precipitation seasonality could leave fingerprints on weighted mean d18O (d18Ow). Current analysis in the manuscript largely ignores changes in d18O itself and only emphases the role of precipitation seasonality but without a quantitate assessment besides correlation analysis. However, the problem is how to decompose these two signal sources rather than simply using correlation analysis. For example, Cai and Tian 2016 used a simple decomposition method to analysis whether precipitation seasonality caused interannual variation in Hong Kong precipitation d18Ow during ENSO years. However, their results indicate that changes in annual d18Ow at Hong Kong during ENSO events are mainly associated with changes in d18O itself rather than precipitation seasonality. Similar decomposition method can be applied in this manuscript to make the question more clearly addressed.

In addition, a parallel analysis on the interannual variations in EASM season d18Ow and NSM season d18Ow or SPR season d18Ow should be performed to reveal the variation of d18O component in specific seasons and its association with interannual d18Ow variation.

*Done. We quantified the effects of seasonal changes in precipitation d18O and precipitation amount on the d18Ow in lines 297-307.*

A potential but fatal risk of the air mass back trajectory analysis in the manuscript is that the analysis only considers air mass movement without considering moisture content. Thus, the analysis result should not be treated as equal to moisture source nor its movement. But when the authors interpreting their back trajectory results, they treated these back trajectories as moisture source trajectories. Further, no information is given on where these air masses picked up or lost water vapor. With this said, the true moisture sources could be totally different from the authors' interpretation in the manuscript. For instance, the C3 ends in the Indian Ocean, but it also travels through the western Pacific region (e.g. the South China Sea). From current results, it is hard to conclude that C3 represents Indian Ocean moisture source.

*Done. Please see Figure 4 and discussion.*

There are several other GNIP stations located within the SPR region and some of them have longer records than Changsha. I am wondering why the authors only used data from Changsha to

address the influence of precipitation seasonality.

GNIP Changsha station is located in the core area of the SPR region.

The use of IsoGSM outputs is too imprudent. In the manuscript, there is no evaluation on the performance of the IsoGSM simulation and no citation of relevant previous evaluations! Does the IsoGSM faithfully re-produce the SPR? Does it correctly simulate the seasonal and interannual precipitation d18O variations in the analyzed region? At least, these questions should be evaluated either from the authors' own analysis or from literature. Otherwise, the results are not solid.

Done. We compared the IsoGSM data with the GNIP data (Supplementary Figure 1).

Overall, the manuscript tends to be descriptive and lacks an in-depth understanding of the controlling mechanism. For example, the interpretation of seasonal precipitation d18O variations as moisture source changes in section 3.2 does not agree with the citation from Baker et al. 2015 in section 4.1 that "the moisture uptake area does not differ significantly between summer and winter".

Done. We have redone the analyses of moisture source.

Why use 1yr-lag correlation? By definition, the d18Ow is not calculated from precipitation amount from the previous or the following year! Thus, the 1yr-lag analysis does not make sense making the analyses and results related to this analysis not scientifically sound.

This was indeed wrong. We have redone the correlation calculations.

The definition of the temporal coverage of seasons or different periods in the manuscript is messy. For example, the authors defined the SPR season for El Nino years as March-to-May between L165-170, but the authors refer to "SPR in March-April during El Nino years (1991-1992)" at L302. Between L165-170, the authors defined SPR and EASM seasons for El Nino years, but what about other years? When the authors analysis "El Nino years (1991-1992)" and "La Nino years (1988-1989)", do you mean Jan 1991 to Dec 1992 for "El Nino years (1991-1992)" and Jan 1988 to Dec 1989 for "La Nino years (1988-1989)"? But both events did not start from Jan or end at Dec. At L345, the MEI is calculated for October-June. Please make all the seasons and periods clear and examine whether the acronyms have a consistent meaning

Done. Lines 140-145.

Figures are hard to read. Please add essential legends to make figures more readable. L24: _50% annual precipitation amount? Similar ambiguity in the main text of the manuscript. Please clarify the difference between the contribution to annual precipitation amount (the weight for calculating annual d18O) and the contribution to annual d18Ow.

Done.

L26: simulated ! please specify

Done.

L29: precipitation d18 ! amount-weight annual precipitation d18O?

Corrected.

L30: Do you mean speleothem d18O records precipitation d18O on the annual scale?

Yes.

L72: d18Op!please define acronym before using; please examining similar problems at other places

Done.

L110-111: But Cai et al. 2018 showed that at least Guilin and Liuzhou is also characterized as significant spring rainfall.

The SPR region was defined by the region ~ 24 °N to 30 °N and 110 °E to 120 °E (Tian and Yasunari, 1998; Wan and Wu, 2007, 2009). Guilin and Liuzhou GNIP stations are located at the edge of the SPR region, while the Changsha station is located in the core area of the SPR region.

L117-118: please provide reference and data to support this conclusion

We now provide references (Yoshimura et al., 2008; Yang et al., 2016) before this sentence.

L154-156: Please show the results for NSM/annual or indicate that NSM/annual equals 1 – EASM/annual.

Yes, we already showed them.

L248: why not using the weighted mean value of EASM and NSM precipitation d18O?

We want to present the absolute value of precipitation d18O, not the weighted mean value.

L250: rainfall amount ! rainfall seasonality?

The whole phrase is "……both rainfall amount and d18Op values during EASM and NSM season……", it means rainfall seasonality.

L305: Why not considering data from other stations? Such as Guilin even has a record longer than that at Changsha.

Changsha station is located in the core area of the SPR region.

Figure 6: Plotting the 1 yr lag time series does not make sense.

Agree, we have redone the correlation calculations.

L345: EASM precipitation amount and NSM precipitation amount: during the following year or the previous year? The MEI is for October-June, but EASM for JJAS and NSM for Oct-May? Why not calculating the contemporary correlation? Even though there is a lead-lag relationship between ENSO and precipitation amount in east Asia, but this is not the scientific question in this manuscript. L385-390: Annual precipitation is mainly from summer monsoon season does not

necessarily mean d18Ow should correlates with precipitation amount. There is no causal relationship between them; one is precipitation seasonality, the other is associated with the "amount effect" on long term scales.

Done, we have redone the correlation calculations.

L389: EASM/annual ! EASM?

Corrected.

[revised manuscript text omitted]

---

## Referee Report (RR1)

Summary: In this manuscript, Zhang et al. investigated the seasonal variability of rainfall amount and precipitation $\delta^{18}O$ over Chinese monson regons, particularly the spring persistent rain (SPR) region. They found the significant contribution (about 50%) to the amount-weighted annual $\delta^{18}O$ from the non-monsoon rains in SPR region. They also suggested that the precipitation seasonality (EASM/NSM), modulated by ENSO, plays a key role in governing the interannual variability of $\delta^{18}O$. I agree with the authors that the spatial variation in precipitation over the MRC is important for the interpretation of $\delta^{18}O$ in Chinese speleothems.

General comments: In general, I feel that this paper is hard to read, especially its abstract, for two reasons. The authors tend to use very complex sentences and too many abbreviations are used (I counted more than 21 abbreviations in total, and some of them are unnecessary). This makes readers extremely difficult (very confusing) to follow the paper, as they have to translate them consistently while reading and very oftern get lost in these abbreviations. Also, the authors lacks the understanding of convection (local convection and regional organized convection) and how these two types of convection affect modern precipitation $\delta^{18}O$. Regional organized convection is largely associated with the monsoons and the local convecton commonly occur in non-monsoon time. Regional organized convection will significantly shift the precipitation isotope to more negative values while local convection has limited impact. The folowing papers are talking about regional and local convecitive activities:

Moerman, J. W., Cobb, K. M., Adkins, J. F., Sodemann, H., Clark, B., & Tuen, A. A. (2013). Diurnal to interannual rainfall $\delta18O$ variations in northern Borneo driven by regional hydrology. Earth and Planetary Science Letters, 369–370(Supplement C), 108–119. https://doi.org/10.1016/j.epsl.2013.03.014

Lekshmy, P. R., Midhun, M., Ramesh, R., & Jani, R. A. (2014). 18O depletion in monsoon rain relates to large scale organized convection rather than the amount of rainfall. Scientific Reports, 4, 5661. https://doi.org/10.1038/srep05661

He, S., Goodkin, N.F., Jackisch, D., Ong, M.R., Samanta, D., 2018. Continuous real-time analysis of the isotopic composition of precipitation during tropical rain events: Insights into tropical convection. Hydrological Processes 32, 1531–1545. https://doi.org/10.1002/hyp.11520

There are several fundamental problems with this manuscript and the authors need to answer them before being considered for publication:

(1). The discussion of the relationship betwee the EASM/NSM ratio and precipitation $\delta^{18}O$ in the SPR region is completely based on IsoGSM simulaiton $\delta^{18}O$ data. The authors used the simulation data from IsoGSM because it replicates the isotope data from GNIP Changsha station in SPR region. If the study is only about this site, the discussion and conclusions are valid. However, their study is about the whole region thus they have to show consistent comparison for multiple stations and the simulation results. I understand that

Changsha station is the only one available in the SPR region with isotope data (five years). However, there are several sations in the regions other than the SPR that have isotope data. The authors should compare the simulation results with isotope data from these stations (at least more than one station) to support or justify why they use the simulated $\delta^{18}O$ in their discussion. Nevertheless, the authors should tune down their conclusions because of the simulated data not the actual observation used in their discussion.

(2). The authors didn't explain the motivation of this study in the Introduction. The title of the manuscript I think obviously suggests that the authors believe the explanation of $\delta^{18}O$ in Chinese speleothems is problematic in previous studies. Therefore, they should clearly state what the problems are with the previous explanation of Chinese speleothems' $\delta^{18}O$. Also, they need to present more clear objectives of this studies and what is the significance of their studuy and implication for paleoclimate studies in this region. In a word, they should rewrite the introduction, specifically the last part of it.

(3). The authors stated "Therefore, we can consider that the $\delta^{18}O_w$ is controlled by both precipitation amount and $\delta^{18}O_p$ during the EASM and NSM seasons in the MRC (lines 291-292)." This is a false or wrong statement. $\delta^{18}O_w$ is calculated based on rainfall amount and $\delta^{18}O_p$ of individual months, and therefore it changes with rainfall amount and $\delta^{18}O_p$. Rainfall amount and $\delta^{18}O_p$ determine the $\delta^{18}O_w$ value, and "control" is not an appropriate word used here. Of course, the factors that control rainfall amount and $\delta^{18}O_p$ control the $\delta^{18}O_w$. Please remove the paragraph between lines 291 and 303. This part is misleading.

(4). ENSO, including El Niño and La Niña, is not a seasonal phenomenon and occur in every few years (3-7 years) not like monsoon which happens each year. ENSO and monsoon operate at different time scales. ENSO is not a driver of seasonal variability in precipitation $\delta^{18}O$ while monsoon is definitely one of the factors that affect the $\delta^{18}O$ seasonal variability. During El Niño event, warm SST area shifts to the eastern pacific regions and supress the convective activities in Asain monsoon region and the draught is often observed. The monsoon is generally weak, and the number and intensity of regional organized convective activities decrease in the Asian region during El Niño while the effect of La Niña is oppisite. Regional organized convection will significantly decrease precipitation $\delta^{18}O$ due to accumulative or integration of convective activity (see the folowing papers):

Gao, J., Masson-Delmotte, V., Risi, C., He, Y., & Yao, T. (2013). What controls precipitation δ18O in the southern Tibetan Plateau at seasonal and intra-seasonal scales? A case study at Lhasa and Nyalam. Tellus B: Chemical and Physical Meteorology, 65(1), 21043. https://doi.org/10.3402/tellusb.v65i0.21043

Vimeux, F., Tremoy, G., Risi, C., & Gallaire, R. (2011). A strong control of the South American SeeSaw on the intra-seasonal variability of the isotopic composition of precipitation in the Bolivian Andes. Earth and Planetary Science Letters, 307(1), 47–58. https://doi.org/10.1016/j.epsl.2011.04.031

Zwart, C., Munksgaard, N. C., Kurita, N., & Bird, M. I. (2016). Stable isotopic signature of Australian monsoon controlled by regional convection. Quaternary Science Reviews, 151(Supplement C), 228–235. https://doi.org/10.1016/j.quascirev.2016.09.010

Local convection has limited impact on precipitation isotope and precipition related to local rain events like events during the spring time in SPR in China generally have high isotope values than those related to regional organized convective activities during the monsoon season.  The seasonality of  EASM/NSM is affected by the Asian monsoon and SPR (likely, SPR is largely associated with local convection and moisture mainly from South China Sea).  Strong (weak) monsoon increases (decreases) the EASM/NSM ratio or the seasonality of ESM/NSM.  When ENSO occurs in any year, it will affect EASM/NSM through affecting the monsoon-related regional convection (frequency and intensity). Weighted average precipitation $\delta^{18}O$ is higher (lower) during El Niño (La Niña) relative to ENSO netral years.

Therefore, the conclusion like the lines 450-452 "We therefore suggest that, over the SPR region the precipitation seasonality (.e.e., the EASM/NSM ratio) modulated by ENSO palys a ke role in governing the internannual variability of $\delta^{18}Op$" is not appropriate. It is also a very confusing statement.  ENSO is a driver of interannual variability of preipitation $\delta^{18}O$, and seasonality of precipitation $\delta^{18}O$ is mainly controlled by the Asian monsoon and SPR in the study area.  When ENSO event occurs, the monsoon and related regional convection will be affected thus the EASM/NSM ratio in that year.

Specific comments: Please see the annotated PDF file attached for line specific comments

[revised manuscript text omitted]

---

## Referee Report (RR2)

Review of Zhang et al. manuscript entitled: Effect of precipitation seasonality on annual oxygen isotopic composition in the area of spring persistent rain in southeastern China and its palaeoclimatic implication

This manuscripts assess the effect of seasonal rain amount and isotopic composition variations in the region of a particular spring persistent rain in China and its influence on the understanding of the isotopic variability observed in regional speleothems. I greatly enjoyed reading this excellent paper that provides a clear outline of the research questions in the introduction and answers those through a through compilation of available data and model results and a careful interpretation. The provided material is novel, comprehensive and the discussion is clear and well structured. All arguments made are based on the observations and consequently the interpretations are sound, new, and highly relevant for both the regional water cycle and paleoclimate studies. The article may well be published as it stands, but I propose subsequently some very minor suggestions, which "possibly" further improve the manuscript.

Minor comments:

**Key points:**

Avoid ACRONYM – for example: * Spring persistent rain region in China reveals isotopic seasonality, which is different from that in other 15 monsoon regions of China.

**Abstract:**

Line 24 (revise): On interannual timescales, less (more) EASM and more (less) NSM precipitation results in higher (lower) amount-weighted annual precipitation $\delta^{18}O$ values. Consequently, higher (lower) speleothems $\delta^{18}O$ values occur during El Niño (La Niña) phases. Moreover, moisture sources and pathways may also impact the mean isotopic composition of rain with respect to the amount ratio of EASM/NSM.

Line 82: replace "Some researchers" with "Huang et al. and Wu et al. "…

Line 101: replace "downloaded" by obtained

Line 106: Link to Ferret – reduce to a high level https://ferret.pmel.noaa.gov/Ferret/ as the download link may evolve

Line 115: replace "not used" with excluded

Line 119: replace "isotope-permitting"

Line 127: replace "reliable" with "consistent"

Line 135: reduce link to higher sublevel: http://www.bom.gov.au/climate/current/

Line 153-154: Sentence on data sources seems misplaced here – it should be placed a few sentences earlier.

Line 168: Link to Figure missing and the order of Figures would change as this Figure would have the number 2 instead of 4: "we calculated the contributions of moisture uptake locations en route to the precipitation in GNIP Changsha station and provide a map "(Figure 4)" showing the percentage of moisture uptake contributing to Changsha precipitation during La Niña (1988-1989) and El Niño (1991-1992) years."

Line 273: I would remove the sentence: The effect of the moisture source on the δ18Op variation will be discussed in section 4.1. As the mentioned section is the next paragraph.

Line 281: Proposal to improve sentence: ...can be calculated from the sum of monthly weighted isotopic values divided by the total amount of precipitation as: The equation would look more scientific if a proper Sum symbol would be used with indices n running from 1-12.

Line 316: … showing rather low δ18Op values … I would prefer replacing "rather" with comparable

Line 370: Typo: dring -> during

---

## Author Response (AR2)

**A point-by-point response to the reviews and a marked revision**

*We are grateful to the editor and reviewers for their constructive comments. We have revised the manuscript accordingly.*

**Referee 1**
Review of Zhang et al. manuscript entitled: Effect of precipitation seasonality on annual oxygen isotopic composition in the area of spring persistent rain in southeastern China and its palaeoclimatic implication
This manuscripts assess the effect of seasonal rain amount and isotopic composition variations in the region of a particular spring persistent rain in China and its influence on the understanding of the isotopic variability observed in regional speleothems. I greatly enjoyed reading this excellent paper that provides a clear outline of the research questions in the introduction and answers those through a through compilation of available data and model results and a careful interpretation. The provided material is novel, comprehensive and the discussion is clear and well structured. All arguments made are based on the observations and consequently the interpretations are sound, new, and highly relevant for both the regional water cycle and paleoclimate studies. The article may well be published as it stands, but I propose subsequently some very minor suggestions, which "possibly" further improve the manuscript.

Minor comments:
**Key points:**
Avoid ACRONYM – for example: * Spring persistent rain region in China reveals isotopic seasonality, which is different from that in other 15 monsoon regions of China.

*Answer# Checked and corrected.*

**Abstract:**
Line 24 (revise): On interannual timescales, less (more) EASM and more (less) NSM precipitation results in higher (lower) amount-weighted annual precipitation $\delta^{18}O$ values. Consequently, higher (lower) speleothems $\delta^{18}O$ values occur during El Niño (La Niña) phases. Moreover, moisture sources and pathways may also impact the mean isotopic composition of rain with respect to the amount ratio of EASM/NSM.

*Answer# Revised.*

Line 82: replace "Some researchers" with "Huang et al. and Wu et al. "…

*Answer# Revised.*

Line 101: replace "downloaded" by obtained

*Answer# Corrected.*

Line 106: Link to Ferret – reduce to a high level https://ferret.pmel.noaa.gov/Ferret/ as the download link may evolve

*Answer# Revised.*

Line 115: replace "not used" with excluded
*Answer# Corrected.*

Line 119: replace "isotope-permitting"

*Answer# Corrected.*

Line 127: replace "reliable" with "consistent"

*Answer# Corrected.*

Line 135: reduce link to higher sublevel: http://www.bom.gov.au/climate/current/

*Answer# Corrected.*

Line 153-154: Sentence on data sources seems misplaced here – it should be placed a few sentences earlier.

*Answer# It should be here.*

Line 168: Link to Figure missing and the order of Figures would change as this Figure would have the number 2 instead of 4: "we calculated the contributions of moisture uptake locations en route to the precipitation in GNIP Changsha station and provide a map "(Figure 4)" showing the percentage of moisture uptake contributing to Changsha precipitation during La Niña (1988-1989) and El Niño (1991-1992) years."

*Answer# Figure 4 should not be put here (data and methods section), it should be in the result section.*

Line 273: I would remove the sentence: The effect of the moisture source on the δ18Op variation will be discussed in section 4.1. As the mentioned section is the next paragraph.

*Answer# Deleted.*

Line 281: Proposal to improve sentence: ...can be calculated from the sum of monthly weighted isotopic values divided by the total amount of precipitation as: The equation would look more scientific if a proper Sum symbol would be used with indices n running from 1-12.

*Answer# Revised.*

Line 316: … showing rather low δ18Op values … I would prefer replacing "rather" with comparable

*Answer# Corrected.*

Line 370: Typo: dring -> during

*Answer# Corrected.*

**Referee 2**

Summary: In this manuscript, Zhang et al. investigated the seasonal variability of rainfall amount and precipitation 18O over Chinese monsoon regions, particularly the spring persistent rain (SPR) region. They found the significant contribution (about 50%) to the amount-weighted annual 18O from the non-monsoon rains in SPR region. They also suggested that the precipitation seasonality (EASM/NSM), modulated by ENSO, plays a key role in governing the interannual variability of 18O. I agree with the authors that the spatial variation in precipitation over the MRC is important for the interpretation of 18O in Chinses speleothems.

General comments: In general, I feel that this paper is hard to read, especially its abstract, for two reasons. The authors tend to use very complex sentences and too many abbreviations are used (I counted more than 21 abbreviations in total, and some of them are unnecessary). This makes readers extremely difficult (very confusing) to follow the paper, as they have to translate them consistently while reading and very often get lost in these abbreviations. Also, the authors lacks the understanding of convection (local convection and regional organized convection) and how these two types of convection affect modern precipitation 18O. Regional organized convection is largely associated with the monsoons and the local convection commonly occur in non-monsoon time. Regional organized convection will significantly shift the precipitation isotope to more negative values while local convection has limited impact. The following papers are talking about regional and local convective activities:

Moerman, J. W., Cobb, K. M., Adkins, J. F.., Sodemann, H., Clark, B., & Tuen, A. A. (2013). Diurnal to interannual rainfall 18O variations in northern Borneo driven by regional hydrology. Earth and Planetary Science Letters, 369-370 (Supplement C), 108-119. https://doi.org/10.1016/j.epsl.2013.03.014

Lekshmy, P. R., Midhun, M., Ramesh, R., & Jani, R. A. (2014). 18O depletion in monsoon rain relates to large scale organized convection rather than the amount of rainfall. Scientific Reports, 4, 5661. https://doi.org/10.1038/srep05661

He, S., Goodkin, N.F., Jackisch, D., Ong, M.R., Samanta, D., 2018. Continuous realtime analysis of the isotopic composition of precipitation during tropical rain events: Insights into tropical convection. Hydrological Processes 32, 1531-1545. https://doi.org/10.1002/hyp.11520

*Answer# Revised.*
*The abstract was reorganized. Some abbreviations (e.g., SEC, $\delta^{18}O_s$) were removed.*
*We agree with that local convection and regional organized convection affect precipitation $\delta^{18}O$. These were discussed in lines 228-234 and in lines 343-347. The suggested references were cited.*

There are several fundamental problems with this manuscript and the authors need to answer them before being considered for publication:

(1) The discussion of the relationship between the EASM/NSM ratio and precipitation 18O in the SPR region is completely based on IsoGSM simulation 18O data. The authors used the simulation data from IsoGSM because it replicates the isotope data from GNIP Changsha station in SPR region. If the study is only about this site, the discussion and conclusions are valid. However, their study is about the whole region thus they have to show consistent comparison for multiple stations and the simulation results. I understand that Changsha station is the only one available in the SPR region with isotope data (five years). However, there are several stations in the regions other than the SPR that have isotope data. The authors should compare the simulation results with isotope data from these stations (at least more than one station) to support or justify why they use the simulated 18O in their discussion. Nevertheless, the authors should tune down their conclusions because of the simulated data not the actual observation used in their discussion.

*Answer# Revised.*

*According to the reviewer's suggestion, we also compare the GNIP data from the Changsha station with the IsoGSM simulation data from the Changsha station and from the key SPR region during 1988-1992 (supplementary figure 1). The high correlation indicates that both precipitation amount and $\delta^{18}O$ data from the IsoGSM simulation are consistent.*

(2) The authors didn't explain the motivation of this study in the Introduction. The title of the manuscript I think obviously suggests that the authors believe the explanation of 18O in Chinese speleothem is problematic in previous studies. Therefore, they should clearly state what the problems are with the previous explanation of Chinese speleothems' 18O. Also, they need to present more clear objectives of this studies and what is the significance of their study and implication for paleoclimate studies in this region. In a word, they should rewrite the introduction, specifically the last part of it.

*Answer# We don't think the title "Effect of precipitation seasonality on annual oxygen isotopic composition in the area of spring persistent rain in southeastern China and its palaeoclimatic implication" indicate the Chinese speleothem $\delta^{18}O$ in previous studies is problematic. In our manuscript, we discuss the precipitation $\delta^{18}O$ changes in the area of spring persistent rain in SE China and its palaeoclimatic implication. The motivation and significance of this study were already illustrated in the introduction in lines 87-101. We said that "The aim of this study is to examine this climate-$\delta^{18}O$ proxy relationship during the instrumental period. To this end, we compare the seasonal variations of precipitation amount and $\delta^{18}O_p$ in the SPR region with other regions of the MRC and discuss the interannual variations of precipitation amount and $\delta^{18}O_p$ in the SPR region and their relationship with the large-scale oceanic-atmospheric circulation.". The debates of Chinese speleothem $\delta^{18}O$ are also discussed in lines 90-95 and in the section "4.3 Implication for paleoclimatic reconstructions".*

(3) The authors stated "Therefore, we can consider that the 18Ow is controlled by both precipitation amount and 18Op during the EASM and NSM seasons in the MRC (lines 291-292)." This is a false or wrong statement. 18Ow is calculated based on rainfall amount and 18Op of individual months, and therefore it changes with rainfall amount and 18Op. Rainfall amount and 18O determine the 18Ow value, and "control" is not an appropriate word used here. Of course, the factors that control rainfall amount and 18Op control the 18Ow. Please remove the paragraph between lines 291-303. This part is misleading.

*Answer# The reviewer may misunderstand this part. As we know, the amount-weighted mean annual precipitation $\delta^{18}O$ ($\delta^{18}O_w$) should be calculated as the following:*
*$\delta^{18}O_w=(P_{Jan}\times\delta^{18}O_{Jan}+P_{Feb}\times\delta^{18}O_{Feb}+\ldots\ldots P_{Dec}\times\delta^{18}O_{Dec})/(P_{Jan}+P_{Feb}+\ldots\ldots P_{Dec})$ Eq. (1)*
*According to the seasonality of precipitation $\delta^{18}O$ (figure 3), however, we simplify the equation (1) in the following mode:*
*$\delta^{18}O_w\approx(P_{EASM-mean}\times\delta^{18}O_{EASM-mean}+P_{NSM-mean}\times\delta^{18}O_{NSM-mean})/(P_{EASM-mean}+P_{NSM-mean})$*
*   $=EASM\%\times\delta^{18}O_{EASM-mean}+NSM\%\times\delta^{18}O_{NSM-mean}$  Eq. (2)*
*Therefore, we said that "we can consider that the $\delta^{18}O_w$ is controlled by both precipitation amount and $\delta^{18}O_p$ during the EASM and NSM seasons in the MRC". We should clarify this in the manuscript.*

(4) ENSO, including El Nino and La Nina, is not a seasonal phenomenon and occur in every few years (3-7 years) note like monsoon which happens each year. ENSO and monsoon operate at different time scales. ENSO is not a drive of seasonal variability in precipitation 18O while monsoon is definitely one of the factors that affect the 18O seasonal variability. During El Nino

event, warm SST area shifts to the eastern pacific regions and suppress the convective activities in Asian monsoon region and the draught is often observed. The monsoon is generally weak, and the number and intensity of regional organized convective activities decrease in the Asian region during El Nino while the effect of La Nina is opposite. Regional organized convection will significantly decrease precipitation 18O due to accumulative or integration of convective activity (see the following papers):

Gao, J., Masson-Delmotte, V., Risi, C., He, Y., & Yao, T. (2013). What controls precipitation δ18O in the southern Tibetan Plateau at seasonal and intra- seasonal scales? A case study at Lhasa and Nyalam. Tellus B: Chemical and Physical Meteorology, 65(1), 21043. https://doi.org/10.3402/tellusb.v65i0.21043

Vimeux, F., Tremoy, G., Risi, C., & Gallaire, R. (2011). A strong control of the South American SeeSaw on the intra-seasonal variability of the isotopic composition of precipitation in the Bolivian Andes. Earth and Planetary Science Letters, 307(1), 47-58. https://doi.org/10.1016/j.epsl.2011.04.031

Zwart, C., Munksgaard, N. C., Kurita, N., & Bird, M. I. (2016). Stable isotopic signature of Australian monsoon controlled by regional convection. Quaternary Science Reviews, 151 (Supplement C), 228-235. https://doi.org/10.1016/j.quascirev.2016.09.010

Local convection has limited impact on precipitation isotope and precipitation related to local rain events like events during the spring time in SPR in China generally have high isotope values than those related to regional organized convective activities during the monsoon season. The seasonality of EASM/NSM is affected by the Asian monsoon and SPR (likely, SPR is largely associated with local convection and moisture mainly from South China Sea). Strong (weak) monsoon increases (decreases) the EASM/NSM ratio or the seasonality of EASM/NSM. When ESNO occurs in any year, it will affect EASM/NSM through affecting the monsoon-related regional convection (frequency and intensity). Weighted average precipitation 18O is higher (lower) during El Nino (La Nina) relative to ENSO netral years.

Therefore, the conclusion like the lines 450-452 "We therefore suggest that, over the SPR region the precipitation seasonality (i.e., the EASM/NSM ratio) modulated by ENSO plays a key role in governing the interannual variability of 18Op" is not appropriate. It is also a very confusing statement. ENSO is a driver of interannual variability of precipitation 18O, and seasonality of precipitation 18O is mainly controlled by the Asian monsoon and SPR in the study area. When ENSO event occurs, the monsoon and related regional convection will be affected thus the EASM/NSM ratio in that year.

*Answer# Revised.*
*We totally agree with the reviewers' comment and suggestion here. We added the related discussion in lines 340-346. The suggested references were cited. In addtion, by using the decomposition method, we separate the influences of precipitation seasonality and monthly $\delta^{18}O_p$ on the difference between precipitation $\delta^{18}O_w$ in El Niño years and La Niña years, the results imply that the difference in $\delta^{18}O_w$ between El Niño and La Niña conditions reflects the differences of both the $\delta^{18}O_p$ and the precipitation seasonality. These were discussed in lines 302-311. Therefore, except for the effect of the seasonal $\delta^{18}O_p$ itself, the seasonal precipitation amount also attributes to the $\delta^{18}O_w$. We also emphasize this in lines 343-347.*

Specific comments: Please see the annotated PDF file attached for line specific comments.

*Answer# Done.*
*The specific comments were addressed accordingly. We two points, which are similar to the comments above, were misunderstood by the reviewer, please see the replies above.*

[revised manuscript text omitted]